# Climate change impacts on the extreme power shortage events of wind-solar supply systems worldwide during 1980−2022

Dongsheng Zheng [1], Dan Tong [1] ✉, Steven J. Davis [2,3], Yue Qin [4], Yang Liu[1], Ruochong Xu[1], Jin Yang[1], Xizhe Yan [1], Guannan Geng [5], Huizheng Che [6] & Qiang Zhang [1]

Economic productivity depends on reliable access to electricity, but the extreme shortage events of variable wind-solar systems may be strongly affected by climate change. Here, hourly reanalysis climatological data are leveraged to examine historical trends in defined extreme shortage events worldwide. We find uptrends in extreme shortage events regardless of their frequency, duration, and intensity since 1980. For instance, duration of extreme low-reliability events worldwide has increased by 4.1 hours (0.392 hours per year on average) between 1980−2000 and 2001−2022. However, such ascending trends are unevenly distributed worldwide, with a greater variability in low- and middle-latitude developing countries. This uptrend in extreme shortage events is driven by extremely low wind speed and solar radiation, particularly compound wind and solar drought, which however are strongly disproportionated. Only average 12.5% change in compound extremely low wind speed and solar radiation events may give rise to over 30% variability in extreme shortage events, despite a mere average 1.0% change in average wind speed and solar radiation. Our findings underline that wind-solar systems will probably suffer from weakened power security if such uptrends persist in a warmer future.

Reliable electricity systems are fundamental protection for energy security, societal sustainability, and national stability[1], which in turn calls for diverse but stable power resources[2]. Solar and wind power would gain places by replacing a large share of traditional fossil-fuel power in the context of achieving the Paris Agreement climate targets[3]. However, these climate-driven energies raise a long-lasting concern about frequent mismatches between supply and demand that often span consecutive hours, days, and in some cases even weeks[4–6]. Hourly mismatches or power gaps with low unmet demands may be easily bridged by strategic designs of multiple technological solutions such as flexible electricity sources (e.g., energy storage) and demand management[7,8]. Yet electricity imbalances characterized by either large unsatisfied demands or day- to even week-long durations—such as the dark doldrum event in Germany[9]—are much more challenging to sustain the stability of wind-solar hybrid electricity system[10].

[1]Ministry of Education Key Laboratory for Earth System Modeling, Department of Earth System Science, Tsinghua University, 100084 Beijing, China. [2]Department of Earth System Science, University of California, Irvine, Irvine, CA 92697, USA. [3]Department of Civil and Environmental Engineering, University of California, Irvine, Irvine, CA 92697, USA. [4]College of Environmental Science and Engineering, Peking University, 100871 Beijing, China. [5]State Key Joint Laboratory of Environment Simulation and Pollution Control, School of Environment, Tsinghua University, 100084 Beijing, China. [6]State Key Laboratory of Severe Weather & Key Laboratory of Atmospheric Chemistry of CMA, Chinese Academy of Meteorological Sciences, 100081 Beijing, China. ✉e-mail: dantong@tsinghua.edu.cn

Great attentions have been paid to average variations in wind and solar resources, underscoring the stability of wind and solar power system would be weakened by climate change[11–17]. A recent published study[18] further exhibited increases in very low photovoltaic power outputs worldwide at the gridded scale, which indicates urgent needs for additional dispatchable backup power and stabilization services to keep generation/demand balance in the future. Although a few studies have provided valuable state[19,20], national[21,22], and regional-specific[23] insights on wind-solar supply shortfalls, the global understanding of the impact of climate change on extreme power shortage events of wind-solar hybrid systems remains largely unclear. Additionally, as the previously limited attention to this emerging issue of electricity supply security, we still lack a uniform analytical framework of climate-change-driven potential extreme shortage events, by combining macro projections of wind-solar generation shares (generally provided by integrated assessment models, IAMs) with hourly mismatches between wind-solar supply and electricity demand. Moreover, there has been still lack of a clear definition and systematic indicators of potential extreme power shortage events in wind-solar electricity systems, which in turn restricts our efforts towards energy security with growing renewable energies under changing climate.

Here, we systematically investigate trends in two types of defined extreme power shortage events (i.e., extreme long-duration events and extreme low-reliability events) of wind-solar systems, their relationships, and the underlying drivers behind the detected trends in 178 countries (see Supplementary Table 1) since 1980. Extreme long-duration events are defined as periods during which electricity demand is unmet for more than 100 consecutive hours according to the upper operation limit of long-duration energy storage[24]; extreme low-reliability events refer to load deficits that last for at least 12 h but have an over 30% power supply gap. We further design three metrics to characterize extreme power shortage events, including frequency (the number of extreme power shortage events in each year), duration (hours of each extreme power shortage event), and intensity (the total power gap in each extreme power shortage event) (Supplementary Fig. 1). Details of extreme power shortage event definitions, data sources, and analytic methods are provided in the Methods section.

In summary, we leverage 43-year (i.e., 1980–2022) hourly Modern-Era Retrospective analysis for Research and Application Version-2 (MERRA-2) reanalysis climatological data to derive area-weighted wind and solar capacity factors (assuming dual-axis solar tracking system) for individual countries. Then, we estimate the hourly electricity supply in wind-solar systems, assuming the reliability-optimized wind/solar generation ratio and the projected install capacity by midcentury. Note that we do not assume a fixed wind-solar share for all the countries, but rather different regional and national wind-solar shares are applied to estimate potential power shortage events for each country, according to the International Energy Agency (IEA; Supplementary Table 2)[25]. The resulting hourly electricity supply and actual/predicted hourly demand from a single recent year are applied to identify two types of defined extreme power shortage events over the past 43 years. Afterwards, we examine the interannual and 20-year interval changes (i.e., two periods of 1980–2000 and 2001–2022) in three metrics of extreme power shortage events and investigate their spatial disparities of fluctuations in the two types of extreme power shortage events. Finally, we combine the two types of extreme power shortage events with climatologic variables to disentangle the potential drivers of such variations particularly for selected 42 major countries (the consideration of electricity demand and regional representation; Supplementary Table 3). Additionally, a series of sensitivity tests on potential constraints such as reanalysis products (i.e., the fifth generation of European Centre for Medium-Range Weather Forecasts (ECMWF) atmospheric reanalysis, ERA5), solar supply technologies (i.e., single-axis solar tracking system and concentrating solar power, CSP), IAMs-based wind-solar ratio and installed capacities, the total wind-solar supply shares (predicted by REgional Model of INvestments and Development (REMIND) model; Supplementary Table 4)[26,27], and decadal analyses (i.e., 1980–1990, 1991–2000, 2001–2010, and 2011–2022) were performed to investigate their impacts on extreme power shortage events.

## Results

### Variability in extreme long-duration shortage events

Figure 1 shows the characteristics of defined extreme long-duration events for wind-solar supply systems across the surveyed 178 countries during the period 1980–2022. Globally, wind-solar supply systems have experienced an increasing trend in extreme long-duration events, although there are repeated up and down undulations irrespective of their frequency, duration, and intensity. Long-duration events worldwide increased significantly at a rate of 0.026 per year in frequency ($P < 0.001$; Fig. 1a), 0.340 h per year in duration (Fig. 1b), and 0.147 per year in intensity (Fig. 1c). Consequently, such increasing trends resulted in considerable variability in extreme long-duration events across the past 43 years. For instance, the annual frequency of extreme long-duration events peaked at 5.8 in 2011, while bottoming at 4.1 in 1983 (-41% difference; Fig. 1d). Such uptrends in extreme long-duration events were also captured by another ERA5 reanalysis data, despite potential underestimates compared to MERRA-2 reanalysis data (Supplementary Note 1 and Supplementary Figs. 2–7).

In particular, we find that extreme long-duration events over the last two decades always outnumbered those over the first two decades (Fig. 1a–c). For example, the annual average duration of extreme long-duration events rose evidently from $146.9 \pm 4.6$ h during 1980–2000 to $155.1 \pm 4.7$ h during 2001–2022 ($P < 0.001$; Fig. 1b). However, extreme long-duration events showed much more complicated decadal changes, with no significant disparities between 1980–1990 and 1991–2000 ($P < 0.05$), but obvious uptrends regardless of its frequency, duration, and intensity between 1991–2000 and 2001–2010 (Supplementary Note 2 and Supplementary Figs. 8–10). We further ranked the annual average frequency, duration, and intensity of extreme long-duration events over the past 43 years (Fig. 1d–f). Our results revealed that the greatest frequency, duration, and intensity of extreme long-duration events occurred intensively in the period between 2001 and 2022, especially the period between 2011 and 2022. For instance, -82% of periods from 2001 to 2022 have been top 20 frequent years of extreme long-duration events (Fig. 1d).

Spatial analyses further illustrated that the observed uptrend in extreme long-duration events is a global-scale phenomenon on a two-decade scale (Fig. 2). This ascending trend in extreme long-duration events existed across all continents (Antarctic is excluded for this study), among which Africa even showed -11 h increase in the duration of extreme long-duration events (Supplementary Fig. 11). However, such upward trends in extreme long-duration events at the global and continental scales mask large spatial disparities across various countries. Indeed, the change in the duration of extreme long-duration events peaked at +74 h in Mozambique, while bottoming at only −34 h in Sao Tome and Principe between 1980–2000 and 2001–2022 (Fig. 2h and Supplementary Fig. 12). Such spatial discrepancy was even more apparent on a decadal scale. For example, duration of extreme shortage events represented the highest increase by +106 h in Chad between 2001–2010 and 2011–2022, but the largest decrease by in Seychelles between 1991–2000 and 2001–2010 (Supplementary Fig. 9). Despite substantial disparities at the national scale, a majority of countries exhibited a growth in extreme long-duration events. In fact, there have been over 70% countries where extreme long-duration events became more frequent, prolonged, and enhanced between 1980–2000 and 2001–2022 (Fig. 2), although such upward trends did not always hold on a decadal scale (Supplementary Fig. 9).

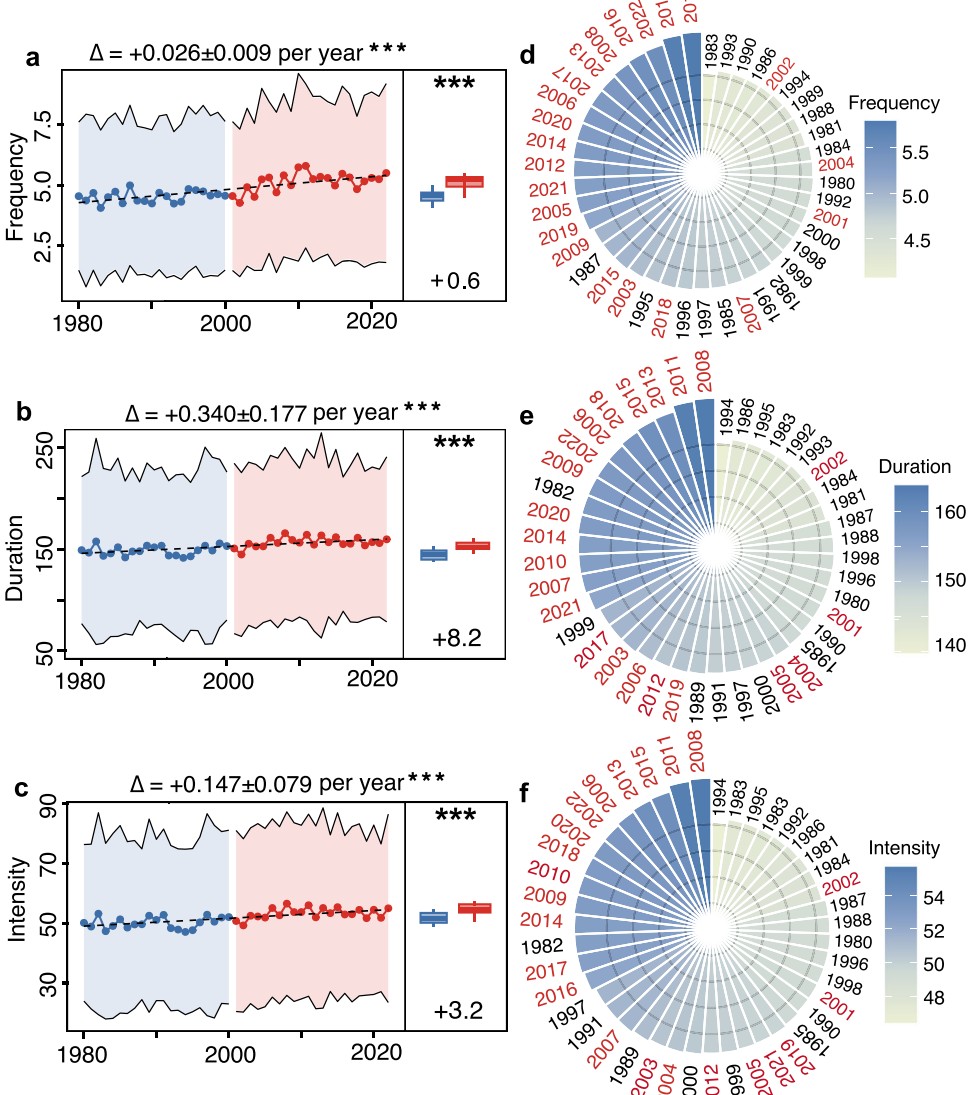

**Fig. 1 | Interannual variability in extreme long-duration events since 1980s.** Interannual changes in the frequency (**a**), duration (**b**), and intensity (**c**) of extreme long-duration events since 1980. The lines and points represent the annual average values of long-duration events and its deviation. The coefficients at the top of the panels indicate robust Theil-Sen's slopes and their deviations, and their corresponding *P* values, examined by Mann–Kendall (MK) test (*n* = 43). The right box-plots denote the difference before 2000 (blue) and after 2000 (red) examined by one-way analysis of variance (ANOVA). *** represents the significance under the level of *P* < 0.001 (*n* = 21 for years during 1980–2000 and *n* = 22 for years during 2001–2022). The boxes and lines indicate the interquartile range and the median, respectively, and whiskers represent the minimum and maximum. Black dashed lines denote linear fitting of annual average values. Rank ordering of the annual average frequency (**d**), duration (**e**), and intensity (**f**) of extreme long-duration events across the surveyed 43 years, in which years between 1980 and 2000 are labeled with black and those between 2001 and 2022 are labeled with red. Source data are provided as a Source Data file.

## Change in extreme low-reliability shortage events

Aside from extreme long-duration events during which electricity demand is unmet for a fairly long period, another challenge for maintaining wind-solar system stability is extreme low-reliability events for which a large amount of flexible energy should be coordinated. There 170 out of 178 countries where extreme low-reliability events have emerged since 1980 (Supplementary Fig. 13), except for countries with comparatively large land areas, such as Russia and China, as a result of their strong spatial aggregations and complementarities[4] (Supplementary Figs. 14–15). Figure 3 shows the change in extreme low-reliability power events between 1980–2000 and 2001–2022. Unfortunately, extreme low-reliability events also followed a significant increasing trend over the past four decades. In the last two decades, extreme low-reliability events have increased by 1.1 (0.069 per year) in frequency (Fig. 3a), 4.1 (0.392 h per year) in

duration (Fig. 3b), and 1.5 (0.131 per year) in intensity (Fig. 3c) relative to the counterpart in the first two decades (*P* < 0.01). This upward trend in low-reliability events was confirmed by the estimates based upon ERA5 data (Supplementary Note 1 and Supplementary Fig. 5).

Similar to extreme long-duration events, despite the global upwards trend in extreme low-reliability events, there are considerable discrepancies across different countries. For instance, the increase in the frequency of extreme low-reliability events reached up to 8.3 in Nigeria, but only grew by 3.3 in New Zealand and even decreased by −2.0 in South Korea on a two-decadal scale (the first column in Fig. 3d). This spatial disparity of variability in frequency of extreme long-duration events would be even larger on a decadal scale, with ranging from nearly +13 in Benin between 1991–2000 and 2001–2010 to only around −12 in Suriname between 2001–2010 and 2011–2022 (Supplementary Fig. 10). However, most countries experienced growth in

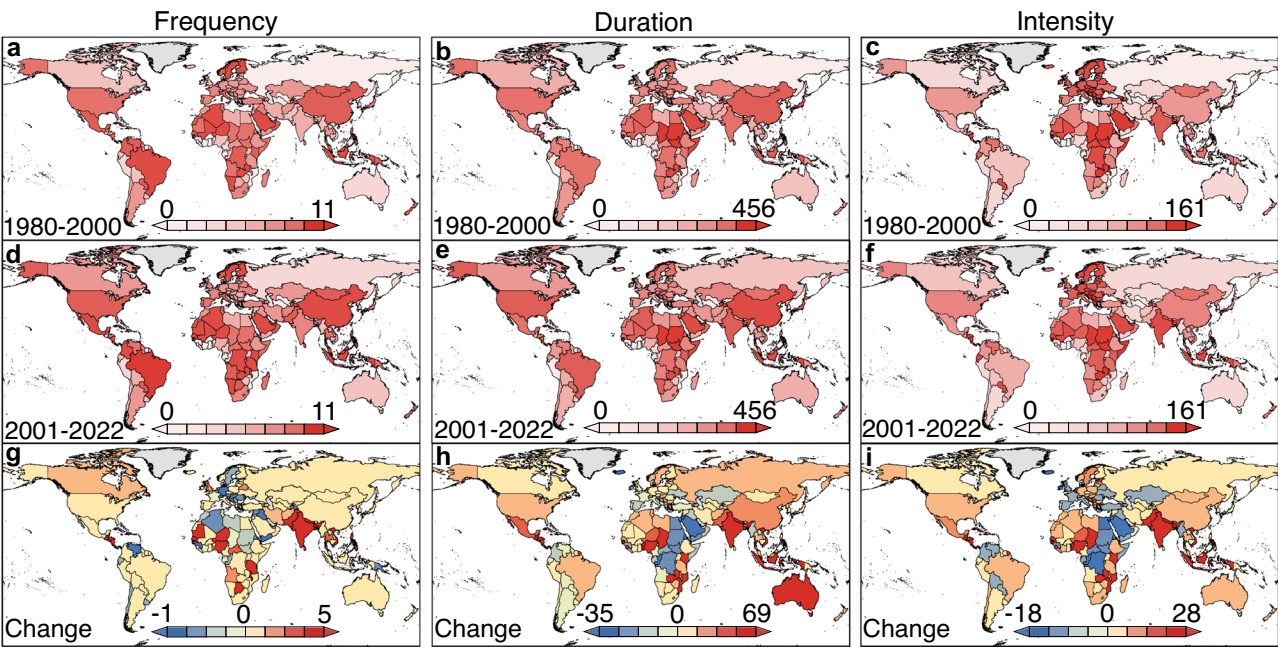

**Fig. 2 | Maps of extreme long-duration events for the surveyed 178 countries during the period 1980–2000 and 2001–2022.** Annual average frequency (**a**), duration (**b**), and intensity (**c**) of extreme long-duration events during the period 1980–2000. Annual average frequency (**d**), duration (**e**), and intensity (**f**) of extreme long-duration events during the period 2001–2022. Changes in annual average frequency (**g**), duration (**h**), and intensity (**i**) of extreme long-duration events between 1980–2000 and 2001–2022. Source data are provided as a Source Data file.

either frequency, duration, and/or intensity of extreme low-reliability events during the period between 1980–2000 and 2000–2022 (Fig. 3), likely attributable to universal reductions in wind and solar electricity generation potential[11,28]. There have been only a few countries, such as France and Turkey, showed declines in all three metrics of extreme low-reliability events (Fig. 3d), ascribed partly to a pronounced decline in very low photovoltaic potential due to reduced cloudy weather and aerosol emissions in these countries[18] (Supplementary Note 3).

### Relationship between extreme long-duration and low-reliability shortage events

Safeguards for the reliability of wind-solar electricity systems are heavily threatened by both extreme long-duration and low-reliability events. Figure 4 further presents the relationship of the two types of extreme power shortage events for 42 major countries (the creteria of major countries and results of 178 countries see Supplementary Note 4 and Supplementary Figs. 16–17), most of which presented the same trends in the two types of extreme power shortage events. Indeed, there are approximately 67% countries where trends in the frequency of extreme long-duration events are in line with those of extreme low-reliability events, particularly in Nigeria that experienced the greatest growth in both extreme low-reliability and long-duration events (upper-right and bottom-left quadrants in Fig. 4a). The estimates based upon another ERA5 data also support the variability characteristics of two types of extreme power shortage events (Supplementary Note 1 and Supplementary Fig. 6). In contrast, the change in extreme low-reliability events is not entirely in accordance with that in extreme long-duration events. For instance, Venezuela exhibited a growth of 2.50 in the frequency of extreme low-reliability events but a decline of −0.70 in the frequency of extreme long-duration events, resulting likely from divergent changes in wind and solar power potentials (Supplementary Fig. 18).

Particularly, changes in extreme power shortage events are unevenly distributed but rather exhibit a latitudinal gradient (Fig. 4). Specifically, high latitudes generally favor slight variabilities in extreme power shortage events, yet low and middle latitudes tend to have drastic changes in extreme power shortage events, attributed possibly to the ongoing weakening of the Hadley circulation associated with human-induced climate warming[29] (Supplementary Note 3). In fact, the greatest changes in extreme long-duration events have emerged in low- and middle-latitude countries (e.g., Nigeria, Peru, Venezuela, Indonesia; blue bubbles at the edge of quadrants in Fig. 4), while high-latitude countries such as Canada and Sweden exhibit a small variability in extreme long-duration events (red rhombuses in the origin of quadrants in Fig. 4).

More importantly, this latitude-dependent change in extreme power shortage events coincides largely with the level of national economic development (Fig. 4). Low- and middle-latitude countries with the highest growth in extreme power shortage events are primarily composed of developing countries, such as India, Indonesia, and Vietnam (Details of the creteria of developing and developed countries see the UN's country classifications[30]). These developing economies are much more vulnerable to extreme power shortage events than developed economies, owing partly to limited dispatchable electricity generation that aids in alleviating temporal mismatches between uncertain supply and demand[31]. Thus, the observed growth in extreme power shortage events is expected to allow more loss-of-load accidents and economic losses in developing economies, thereby enlarging the existing disparities in electricity system reliability between developing and developed countries[31].

### Drivers of growing extreme shortage events

We explore climatological drivers underlying the detected growing extreme power shortage events by combining extremely low wind speed and solar radiation that are defined as below the 10th percentile of the daily average value across 43 surveyed years (Fig. 5). We find that, the growth in extreme shortage events is attributable to prolonged extremely low wind speed and solar radiation (Fig. 5a–c), which also conforms with significant declines in the annual mean wind speed and solar radiation since the 1980s ($P < 0.05$; Supplementary Figs. 19–22 and Supplementary Note 3). Although variabilities in extreme power shortage events are controlled by both wind and solar

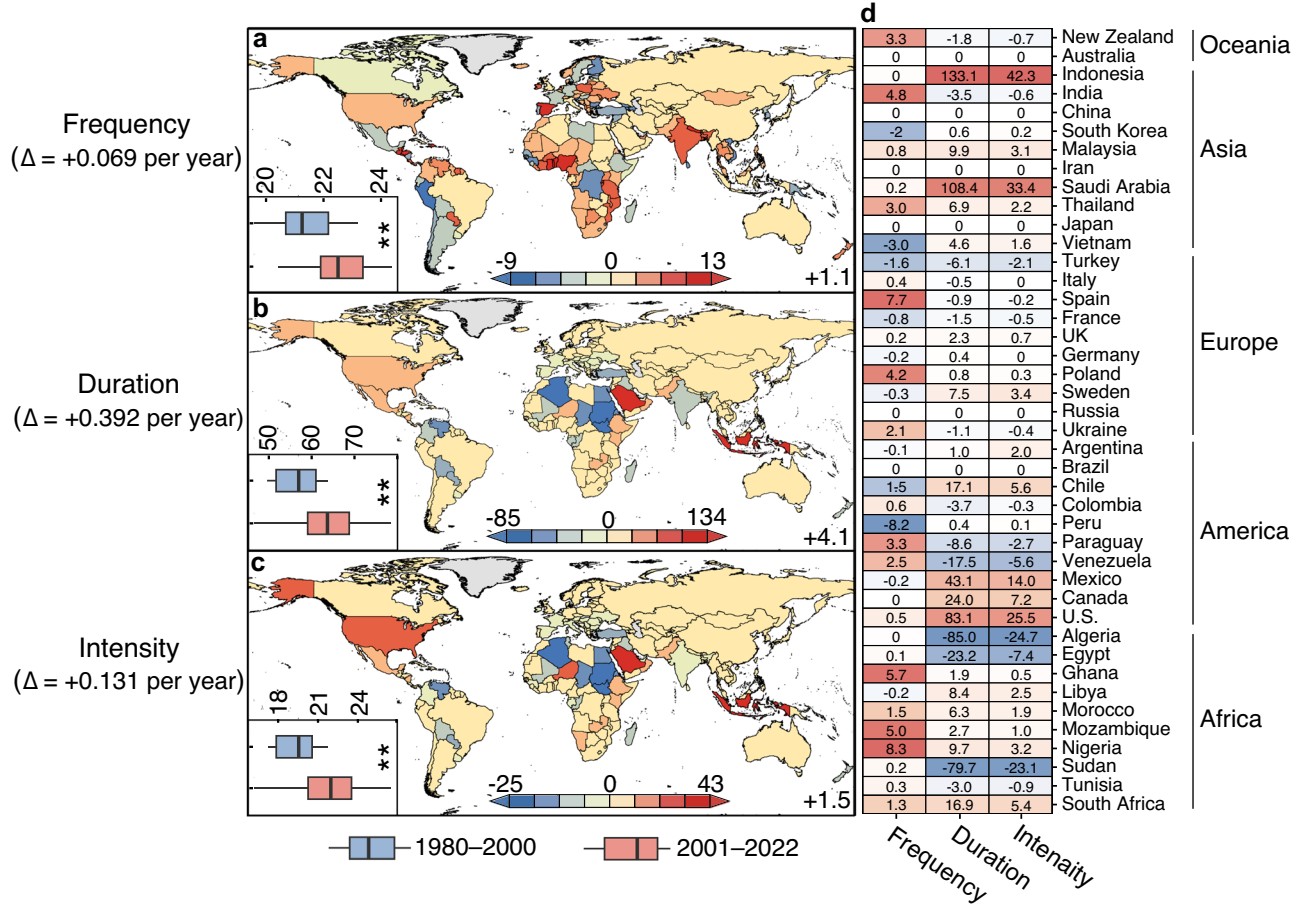

**Fig. 3 | Changes in extreme low-reliability events since 1980.** Changes in annual average frequency (**a**), duration (**b**), and intensity (**c**) of extreme low-reliability events between 1980–2000 and 2001–2022. The left-bottom boxplots denote the difference between 1980–2000 (blue) and 2000–2022 (red) examined by one-way analysis of variance (ANOVA), where ** represent the significance under the level of $P < 0.01$ ($n = 21$ for years during 1980–2000 and $n = 22$ for years during 2001–2022).

The boxes and lines indicate the interquartile range and the median, respectively, and whiskers represent the minimum and maximum. **d** Changes in extreme low-reliability events for 42 major countries between 1980–2000 and 2001–2022. The detailed creteria to select major countries is provided in Supplementary Note 4. Source data are provided as a Source Data file.

energies, their relative importance are different and are partly affected by wind-solar ratio and installed capacities. The changes in extreme power shortage events are primarily ascribed by the variability in wind power in wind-heavy system (Supplementary Fig. 23), while additional analyses using IAMs-based wind/solar ratio (a solar-heavy system) and install capacities reveal that wind power does not always place a dominant position in extreme power shortage events (Supplementary Notes 5–6 and Supplementary Figs. 24–29).

Moreover, we further define a compound extremely wind speed and solar radiation event, a period during which both wind speed and solar radiation are below the 10th percentile of the daily average value across 43 surveyed years[20,32,33]. We find that most countries have experienced increasing defined compound extremely low wind speed and solar radiation. Indeed, there are over 70% countries with an uptrend in annual hours of compound extremely low wind speed and solar radiation events, although some countries (e.g., Peru, Algeria, and France) exhibit the decline in compound extremely low wind speed and solar radiation between 1980–2000 and 2001–2022 (Supplementary Note 7 and Supplementary Fig. 30). More importantly, such a growing trend in compound extremely low wind speed and solar radiation events is in largely accordance with the observed increase in extreme long-duration and low-reliability shortage events since the 1980s (Supplementary Fig. 31). For instance, a higher intensity of long-duration events in the recent period (e.g., 2010, 2016, and 2022) has been in line with a longer compound extremely low wind

speed or solar radiation events, while a lower intensity of long-duration events before 2000 (e.g., 1983 and 1998) is consistent with a shorter compound extremely low wind speed or solar radiation.

Notably, there are large disproportionalities between changes in extreme power shortage events and climatological variables (e.g., compound extremely wind speed and solar radiation events and average wind speed and solar radiation; Fig. 5d, e). Small changes in annual average wind speed and average solar radiation generally indicate considerable changes in compound extremely wind speed and solar radiation events. Indeed, although the change in annual average wind speed and average solar radiation is as low as 1.0% (the pink circle in Fig. 5d), the annual hours of compound extremely low wind speed and solar radiation events change by more than 12.5% (the blue circle in Fig. 5d, e). More importantly, small changes in compound extremely low wind speed and solar radiation events usually suggest substantial changes in extreme power shortage events. In fact, a mere 12.5% change in the annual hours of compound extremely low wind speed and solar radiation events may drive an average 30.4% change in extreme long-duration and low-reliability events (the red circle in Fig. 5e). The results were evidenced by the results using ERA5 reanalysis data (Supplementary Note 1 and Supplementary Fig. 7). The disproportionalities of changes between extreme power shortage events and climatological variables highlight that even mild climate changes are expected to pose a severe challenge to the security of wind-solar generation systems.

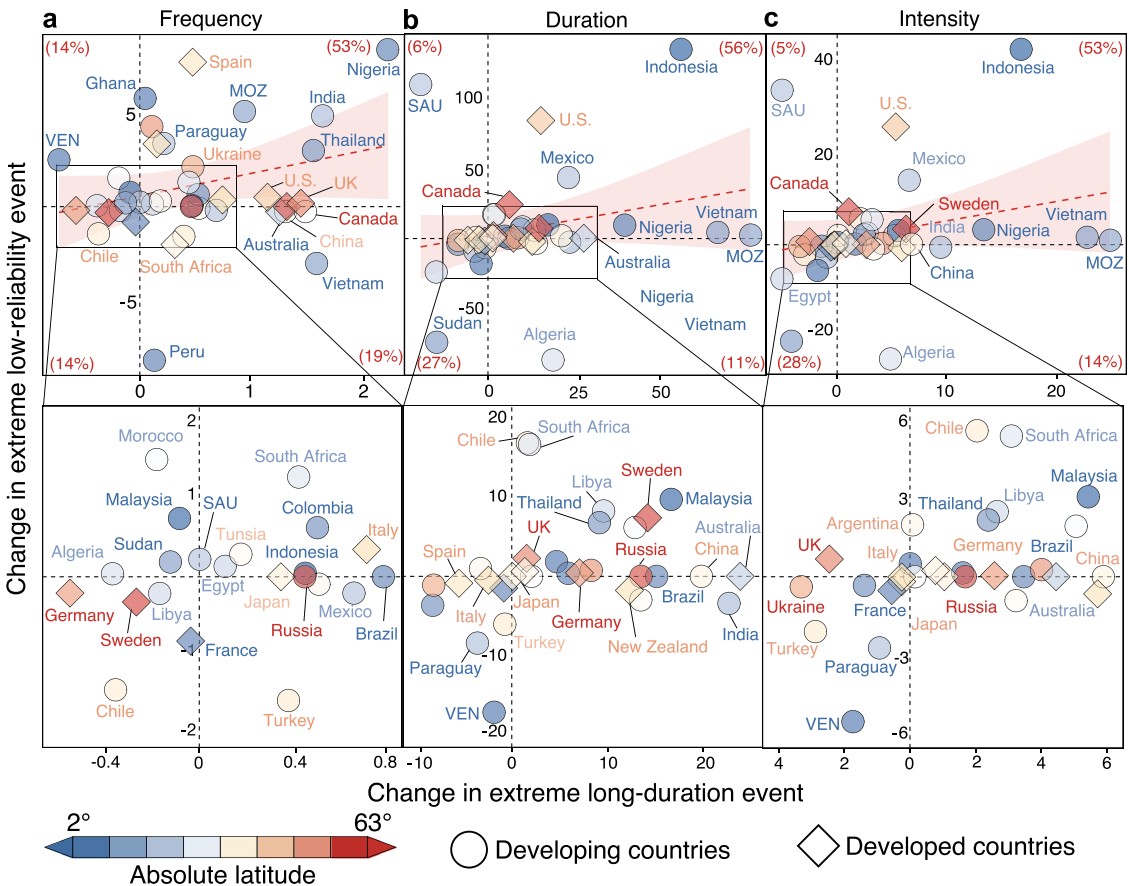

**Fig. 4 | Relationship of changes in extreme long-duration and low-reliability events at the national scale.** Changes in frequency (**a**), duration (**b**), and intensity (**c**) of extreme long-duration and low-reliability events for 42 major countries between 1980–2000 and 2001–2022. Red digits represent the proportion of countries that fall in the quadrant. Dashed red lines with shallow red shading are linear fittings between the two types of events and their 95% confidence intervals. SAU, MOZ and VEN denote Saudi Arabia, Mozambique, and Venezuela, respectively. Source data are provided as a Source Data file.

## Discussion

This study defines three metrics—frequency, duration, and intensity—to examine the interannual variabilities in potential extreme power shortage events of wind-solar hybrid systems since 1980. Our results suggest that the past several decades may have witnessed increasingly frequent, prolonged, and intensified extreme shortage events across the globe, particularly in low- and middle-latitude developing countries. Such ascending trends in extreme power shortage events are attributable to growth in extremely low wind speed and solar radiation, affected partly by increased cloud cover[18], reduced aerosol emissions[34,35], enhanced surface roughness[36,37], and weakened pressure gradient[38–40] associated with global warming (Supplementary Note 3). More importantly, if the detected growing extreme power shortage events persist in a warmer future, wind-solar systems may face weakened energy security and enhanced system costs.

Our estimates are restricted by several potential limitations and uncertainties like meteorological data and technology assumption. In terms of meteorological data, on one hand, current hour-scale resolution may not capture the dramatical change in wind speed at the minute scale. It limits our refined knowledge on extreme power shortage events, which in turn calls for future access to high-resolution meteorological data (Supplementary Note 8). On the other hand, the estimates of system reliability may be sensitive to reanalysis data choice[4], therefore, we have added ERA5 data to verify our main results (Supplementary Note 1). It shows that the estimates based upon ERA5 data are largely in accordance with those using MERRA-2 (Supplementary Figs. 3–7), despite a slight underestimate in both long-duration and low-reliability power shortage events (Supplementary

Fig. 2). As for technology assumption when estimating the solar capacity factors, our analyses based upon dual-axis tracking system may result in potential uncertainty of system reliability and power shortage events, as the continued progress of solar tracking systems. Our tests with consideration of single-axis tracking type and tracking system progress show lower system reliabilities but similar increasing trends in both long-duration and low reliability events of wind-solar supply systems (Supplementary Note 9 and Supplementary Figs. 32–36). Besides, CSP, as a potential important technology towards the reliability of solar supply system[41], is also considered into our technology sensitivity tests according to its future supply share (Supplementary Table 5), which shows the consistent findings (see Supplementary Note 10 and Supplementary Figs. 37–41).

Our study may also be affected by wind-solar supply parameter settings and demand profile. To be specific, we estimated wind-solar ratio and installed capacities based on the reliability-optimized method, which likely in turn influences our results about power shortage events and their driving forces. Our extended sensitivity tests with IAMs-based wind and solar proportion and install capacities do not alter the uptrend in power shortage events, while these tests indicate that the importance of wind and solar energies would differ from various extreme shortage events and their metrics (Supplementary Notes 5–6 and Supplementary Figs. 24–29). Besides, we assume an electricity system with a certain share of wind-solar energy by mid-century around the world (Supplementary Table 2)[25]; however, the future share of wind-solar power likely varies and is even more aggressive for individual countries[42]. Our additional sensitivity tests, which assumed 80% and 100% wind-solar hybrid systems globally,

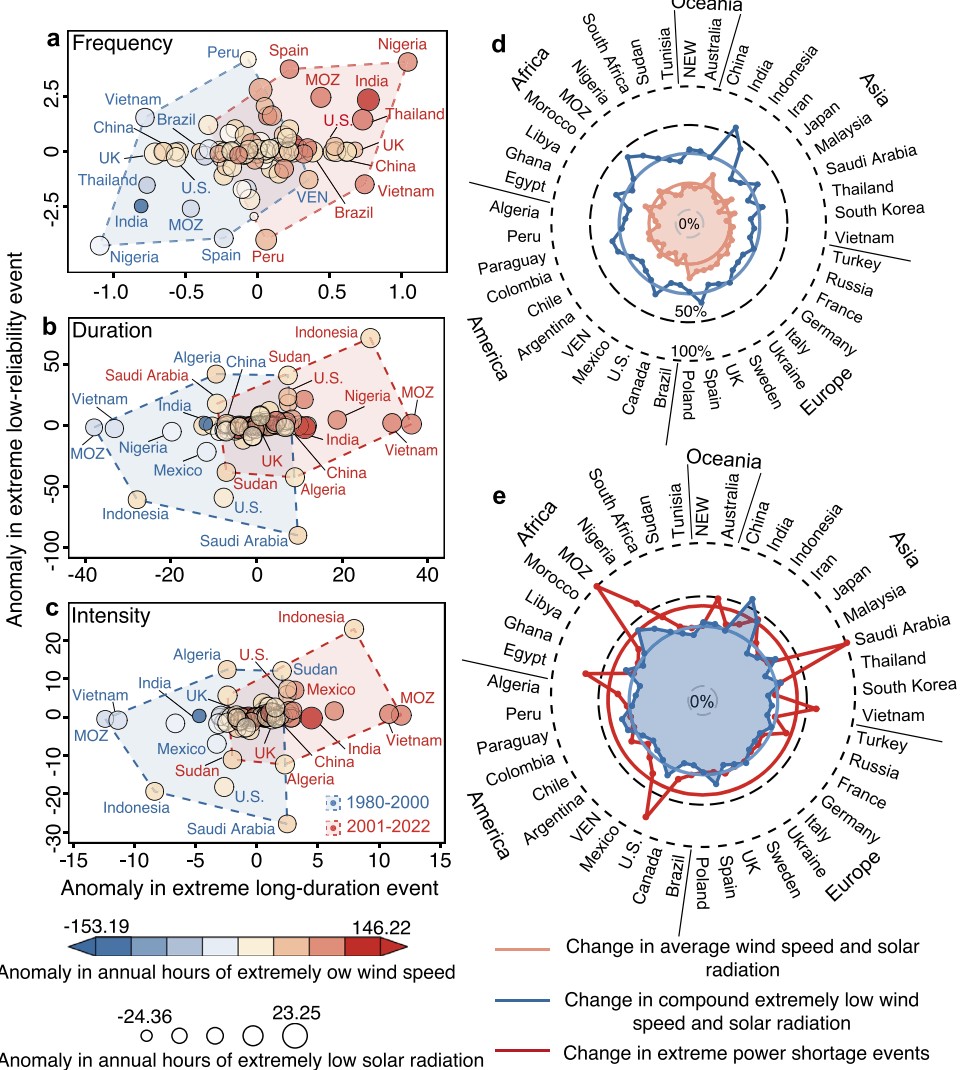

**Fig. 5 | Changes in extreme power shortage events with extremely low wind speed and solar radiation.** Relationship between anomalies in frequency (**a**), duration (**b**), and intensity (**c**) of extreme power shortage events and anomalies in annual hours of extremely low solar radiation and wind speed at hub height for 42 major countries. **d** Relationship between change in annual average wind speed and average solar radiation and change in annual hours of compound extremely low wind speed and solar radiation events between 1980–2000 and 2001–2022.

**e** Relationship between the relative change in annual hours of compound extremely low wind speed and solar radiation events and the relative change in extreme power shortage events between 1980–2000 and 2001–2022. The relative change above 100% is visualized as 100% change. MOZ and VEN denote Mozambique and Venezuela, respectively. The lines and circles denote individual country's changes and the average changes across 42 major countries, respectively. Source data are provided as a Source Data file.

even exhibited a greater intensity of potential extreme power shortage events (Supplementary Figs. 42–43). Moreover, another sensitivity test of wind-solar share in 2050 given potential importance of economics and renewable policies (1.5 °C climate target from the REMIND model[27,43]) was performed, and the results do not change the robustness of our main findings concerning to growing extreme power shortage events and its driving forces (Supplementary Note 11 and Supplementary Figs. 44–48). In addition, the future electricity demand profile may be quite distinct from the current load, which would cause potential underestimates or overestimates of extreme power shortage events. Therefore, we further utilized the future demand profile by mid-century[44] to examine trends in extreme power shortage events, showing that our results are not specifically dependent on demand patterns (Supplementary Fig. 49).

Despite these potential uncertainties in our estimates, our findings have important implications for policy makers and future research. Even though the reliability-optimized wind-solar system is capable of satisfying 70–90% of the power demand across the world regardless of country's land areas and latitudes (Supplementary Fig. 50), potential extreme power shortage events create considerable challenges to electricity system stabilization and energy security in the future. Unfortunately, this existing security risks are very likely aggravated when considering that even mild future climate changes may drive a drastic increase in extreme long-duration and low-reliability power shortage events of wind-solar systems.

Towards the security of future electricity systems, long-duration and low-reliability power shortage events would have unequal risks and may call for different scales and types of flexible energy to help smooth fluctuations, whose challenges in turn likely depends on the future storage technology development and the scale of stable backup fossil-fuel-based power capacity[19,20,25]. For example, if cost-effective long-duration storage technology matures and limited backup fossil capacity is kept in the future, the threat of low-reliability events

probably outweighs that of long-duration ones. On the contrary, long-duration events may be a severer threat to power systems than low-reliability ones. On top of that the combination of these two types defined events probably bring much more serious risks than either type of event. Long-duration events tend to trigger electricity outage accidents which last for a long period, while low-reliability events likely lead to short but large-scale supply failure. Unfortunately, our results demonstrated that both show an increasing trend, suggesting that future electricity system probably experience both longer and higher power gaps in the context of lasting climate change. The combined risk may call for a more urgent need to backup electricity generation, and thus strategic integration of multiple flexibility resources would be a feasible solution to address such electricity security problems in the future.

Furthermore, the detected growths in both extreme long-duration and low-reliability power shortage events require a mass of flexible backup electricity generation to bridge power supply/demand mismatches, which could be provided via energy storage[45], transmission network expansion[46,47], overbuilding[5], and fossil-fuel generators[48]; however, these excess devices to meet extreme power shortage events would in turn lead to substantial increases in investment and maintenance costs when operating at a relatively low capacity. Moreover, these technological solutions would create increased complexity in balancing technical reliability and economic feasibility in a high-penetration wind-solar hybrid electricity system. Although a strategic combination of multiple promising technologies may provide possible solutions, it still allows much more complicated energy planning and grid-stabilization management across the world[49,50].

More importantly, growth in extreme power shortage events is unevenly distributed across the world, but rather larger variabilities have emerged in low- and middle-latitude developing countries[29]. However, these developing economies tend to be more vulnerable for warming-induced power load shortfalls, which can be ascribed to their relatively poor energy infrastructure and grid-stabilization capacity[31]. As a consequence, the observed increases in extreme power shortage events will likely cause more severe outage accidents and higher socioeconomic costs in developing economies. Therefore, the growth in extreme power shortage events probably enlarges potential unequal burdens in terms of energy security between developed and developing countries.

## Methods
### Climatological variables
We obtained hourly climatological variables from MERRA-2 reanalysis product (1980–2022), with a horizontal resolution of 0.5° by latitude × 0.625° by longitude (totally 207,936 pixels across the world)[51]. The downloaded climatological variables, including 10-meter eastward wind (m s$^{-1}$) (variable name: U10M), 10-meter northward wind (m s$^{-1}$) (variable name: V10M), eastward wind at 50 meters (m s$^{-1}$) (variable name: U50M), northward wind at 50 meters (m s$^{-1}$) (variable name: V50M), air density at surface (kg m$^{-3}$) (variable name: RHOA), surface incoming shortwave flux (W m$^{-2}$) (variable name: SWGDN), top atmosphere incoming shortwave flux (W m$^{-2}$) (variable name: SWTDN), 2-meter air temperature (K) (variable name: T2M), 2-meter eastward wind (m s$^{-1}$) (variable name: U2M), as well as 2-meter northward wind (m s$^{-1}$) (variable name: V2M), are served to calculate hourly wind and solar capacity factors at each specific grid cells worldwide.

### Wind and solar capacity factors
Capacity factor that describes wind and solar energy resource potentials at a given location is defined as the ratio of actual output capacity to rated nameplate capacity[50]. Wind capacity factor is estimated using the power curve of standard wind turbine that depicts the actual electricity output as a function of wind speed[52] as Eq. (1).

$$CF_w = f_w(V_{100c}) \tag{1}$$

where $CF_w$ and $f_w$ represent wind capacity factor and the power curve of standard wind turbine, respectively. We assume the GE 2.5 MW wind turbine (https://en.wind-turbine-models.com/turbines/1293-ge-general-electric-ge-2.5-103) (Supplementary Fig. 51) that has been widely deployed for wind energy development worldwide[52]. $V_{100c}$ denotes the wind speed at standard air density at hub height, and thus the wind speed at 10 m and 50 m height should be extrapolated to 100 m above ground level using the power-law relationship[4] as Eqs. (2) and (3):

$$V_{100} = V_{10} \times (100/10)^a \tag{2}$$

$$a = \log(V_{50}/V_{10}) / \log(50/10) \tag{3}$$

Where $\alpha$ refers to wind profile exponent. $V10$, $V50$, and $V100$ are the actual wind speed at 10 m, 50 m, and 100 m above ground level, respectively.

The actual wind speed should be further corrected to the wind speed at standard air density for the power curve of wind turbine, since wind power potential increases linearly with air density[53]. The adjusted equation[53] is described as Eq. (4):

$$V_{100c} = V_{100} \left( \frac{\rho_1}{\rho} \right)^{1/3} \tag{4}$$

where $V_{100c}$ refers to hub height wind speed at standard air density; $\rho_1$ and $\rho$ indicate actual air density (RHOA) and standard air density (1.225 kg m$^{-3}$), respectively.

With regard to solar capacity factor, we assume that utility-scale photovoltaic systems are deployed for solar power generation. Solar capacity factor depends largely on in-panel solar radiation and operational temperature[54] as Eq. (5):

$$CF_s = \frac{P_a}{P_r} = \frac{I_\Sigma \times EF \times TEM_{coef}}{I_{std} \times EF} = \frac{I_\Sigma \times TEM_{coef}}{I_{std}} \tag{5}$$

where $CF_s$ denotes solar capacity factor; $P_a$ and $P_r$ refers to hourly actual power generation and the rated power generation per unit land area, respectively; $I_\Sigma$ is solar radiation intercepted by photovoltaic module (kWh m$^{-2}$), and $I_{std}$ indicates hourly power generation per unit land area under standard test conditions (with an operational temperature of 25 °C, $I_\Sigma = 1000$ kWh m$^{-2}$, and an air mass of 1.5 spectrum); $EF$ and $TEM_{coef}$ represent the conversion efficiency of in-panel solar radiation to electricity and temperature-adjusted coefficient, respectively.

$TEM_{coef}$ is a function of actual photovoltaic cell temperature ($T_{cell}$) that relies largely on irradiation, ambient temperature, as well as surface wind speed[12] as Eqs. (6) and (7):

$$TEM_{coef} = 1 + \lambda \times (T_{cell} - T_{STC}) \tag{6}$$

$$T_{cell} = c_1 + c_2 \times T_{amb} + c_3 \times I_\Sigma + c_4 \times V_2 \tag{7}$$

where $\lambda$ denotes the typical temperature response efficiency for monocrystalline silicon modules (−0.005 °C$^{-1}$); $T_{STC}$ is photovoltaic cell temperature under standard test condition (25 °C), and $T_{cell}$ represents the actual photovoltaic cell temperature (°C); $c_1$ is a constant (4.3 °C); $c_2$, $c_3$, and $c_4$ refer to temperature coefficient (0.943), irradiation coefficient (0.028 °C m$^{-2}$ W$^{-1}$), and wind speed

coefficient ($-1.528\,°C\,m^{-1}$), respectively; $T_{amb}$ and $V_2$ indicate ambient temperature at 2-meter height and wind speed at 2 m above ground level, respectively.

We separated in-panel solar radiation ($I_\Sigma$) into direct beam radiation ($I_{B\Sigma}$), diffuse radiation ($I_{D\Sigma}$), and reflected radiation ($I_{R\Sigma}$) received by photovoltaic modules[54] as Eq. (8):

$$I_\Sigma = I_{B\Sigma} + I_{D\Sigma} + I_{R\Sigma}$$
$$= I_{BH} \times \cos\theta_0 + I_{DH} \times R_d + I_H \times \frac{\rho(1-\cos\Sigma)}{2} \quad (8)$$

where $I_{B\Sigma}$ is calculated using direct beam radiation on horizonal surface ($I_{BH}$) and solar incidence angle ($\theta_O$). The $\theta_O$ is estimated by taking into account solar altitude angle, solar azimuth angle, solar hour angle, solar panel azimuth angle, and solar panel tilt angle based upon dual-axis tracking system[55]. we estimated $I_{D\Sigma}$ through horizontal surface diffuse radiation ($I_{DH}$) and its conversion coefficient ($R_d$) that depends largely on solar azimuth angle, solar zenith angle, sky brightness, as well as sky clearness[56]; $I_{R\Sigma}$ is calculated using surface incoming shortwave flux ($I_H$, namely SWGDN), solar panel tilt angle ($\Sigma$), and surface albedo ($\rho$) that is generally set to 0.2, a default value for ground or grass. $I_{BH}$ and $I_{DH}$ are quantified through separating $I_H$ through an empirical piecewise equation[57] that describes the fraction of $I_{DH}$ to $I_H$ using solar altitude angle, SWGDN, and SWTDN.

## Hourly electricity demand

We obtained hourly electricity demand data at country level using Tong et al. method[4] that integrates datasets derived from public power system datasets, previous studies, as well as government and electricity market websites. With regard to countries unavailable on the real-world demand data, we introduced Toktarova et al. hourly electricity demand[44] dataset that was projected using actual demand load profiles, socio-economic variables, as well as climatological factors. The resulting single-year electricity demand profile was joined 43 times to generate a continuous hourly demand curve during 1980–2022.

## Installed capacities and hourly supply of wind and solar powers

The installed capacity of wind and solar for individual countries are calculated based upon electricity demand and the future projected share of wind and solar[4] as Eqs. (9) and (10):

$$Capacity_w = Dem_{avg} \times Share_{sw} \times Ratio_w / CF_{wm} \quad (9)$$

$$Capacity_s = Dem_{avg} \times Share_{sw} \times (1 - Ratio_w) / CF_{sm} \quad (10)$$

Where, $Capacity_w$ and $Capacity_s$ denote the installed capacities of wind and solar, respectively. $DEM_{avg}$ is average hourly electricity demand. $Share_{sw}$ represents the predicted share of wind and solar to the total electricity generation by mid-century according to IEA (Supplementary Table 2)[25]. The IEA predicts future wind-solar share of each region and major countries (e.g., United States, India, and China), yet the shares of wind-solar power in small countries are unavailable. We therefore assigned regional wind-solar shares to countries without prediction. $Ratio_w$ and $1$-$Ratio_w$ are the optimal ratio of wind and solar energy to wind-solar systems (Supplementary Fig. 50), respectively, which is described in Supplementary Note 12 in detail. $CF_{wm}$ and $CF_{sm}$ indicate area-weighted average wind and solar capacity factors during the study period through 1980–2022, respectively.

The resulting installed capacities of wind and solar powers are further applied to calculate hourly wind and solar supply by multiplying hourly wind and solar capacity factor as Eqs. (11) and (12):

$$Sup_{w,t} = Capacity_w \times CF_{w,t} \quad (11)$$

$$Sup_{s,t} = Capacity_s \times CF_{s,t} \quad (12)$$

Where, $Sup_{w,t}$ and $Sup_{s,t}$ represent wind and solar electricity supply at hour $t$, respectively. $CF_{w,t}$ and $CF_{s,t}$ refer to the area-weighted average capacity factors of wind and solar power at hour $t$, respectively.

## Extreme power shortage events in wind-solar supply system

We have investigated changes in two types of extreme shortage events: extreme long-duration events and extreme low-reliability events. The former is defined as a period during which unsatisfied electricity demand lasts for at least consecutive 100 h. In general, the 100-h is the upper limit of discharge duration of long-duration energy storage techniques that tend to act as the primary dispatchable generation sources to fill the mismatches between uncertain renewable supply and demand[24,46]. Another type of events is extreme low-reliability events that persist only a short time (more than 12 h) but encompass over 30% power gap, a well-recognized challenge to the security of electricity system.

The two types of defined extreme shortage events are characterized by three characteristics: frequency (the number of extreme shortage events in each year), duration (hours of each event), and intensity (the total power gap over each event). We take the Germany electricity load in 1981 as an example to illustrate the three indices. Germany had experienced potential four extreme long-duration shortage events in 1981 (frequency = 4), and one of these events had emerged over the period from 15:30 on Jan. 26 to 08:30 on Feb. 1st (duration = 139 h) and had showed the total power gap of 50.76 (intensity = 50.76) (Supplementary Fig. 1).

The calculation of extreme power shortage events and their three indices rely on the reliability and power gap of wind-solar hybrid systems. The reliability is the share of demand that wind-solar electricity system can satisfy, and the remaining unmet demand represents power gap as Eqs. (13)–(15):

$$Relib_t = \begin{cases} Sup_t/Dem_t \times 100\%, & Sup_t < Dem \\ 1, & Sup_t > Dem_t \end{cases} \quad (13)$$

$$Sup_t = Sup_{w,t} + Sup_{s,t} + Dem_t \times (1 - Share_{sw}) \quad (14)$$

$$Gap_t = \begin{cases} (1 - Relib_t) \times 100\%, & Relib_t < 1 \\ 0, & Relib_t = 1 \end{cases} \quad (15)$$

where $Relib_t$ and $Gap_t$ represent the reliability and power gap of wind-solar power system at hour $t$, respectively; $Sup_t$ and $Dem_t$ are electricity supply and electricity demand at hour $t$, respectively; $CF_{w,t}$ and $CF_{s,t}$ indicate area-weighted wind and solar capacity factors at hour $t$, respectively.

## Statistical analysis

We examined the linear trend in power shortage events, capacity factors, and climatic factors using Theil-Sen's slope estimator, a classical non-parametric method to detect interannual variation[58,59]. We further applied Mann–Kendall test to examine the statistical significance of Theil-Sen's slope[60]. The observed decadal changes in power shortage events, capacity factors, and climatic factors were examined using one-way analysis of variance (ANOVA) that is a widely used method to detect statistically significant differences between the mean values of various groups[61]. The P value, which describes the probability under assumption of null hypothesis[59,60], is applied to determine the

significance of Mann–Kendall test and ANOVA. All the statistical analyses and significance detection were performed using R (version 4.3.1) (R Foundation for Statistical Computing) with software packages.

## Reporting summary

Further information on research design is available in the Nature Portfolio Reporting Summary linked to this article.

## Data availability

Hourly reanalysis climatological fata for the calculation of wind and solar capacity factors can be accessed from MERRA-2 and ERA5 (https://cds.climate.copernicus.eu/cdsapp#!/home). Future supply share of CSP and photovoltaic solar are available from International Institute for Applied Systems Analysis (IIASA; https://data.ece.iiasa.ac.at/ar6/). Raw data for extreme power shortage events for individual countries is provided on Zenodo (https://zenodo.org/records/11066289). Source data are provided with this paper.

## Code availability

The code for collecting ERA5 climatic reanalysis data can be acquired from CDS Toolbox (https://cds.climate.copernicus.eu/toolbox/doc/index.html). The code for estimating hourly wind and solar power capacity factors used on reanalysis data is available on https://github.com/carnegie/Create_Wind_and_Solar_Resource_Files.

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

## Acknowledgements

This work was supported by National Natural Science Foundation of China (grant nos. 72274106 and 42222507), Tsinghua University Initiative Scientific Research Program (grant nos. 20223080041), International Joint Mission on Climate Change and Carbon Neutrality, and China Meteorological Administration "Research on Value realization of climate ecological products" Youth Innovation Team Project (No. CMA2024QN15).

## Author contributions

D.T. and D.Z. designed the study. D.Z. performed the analyses, with support from D.T., Q.Z., S.J.D., H.C., G.G., and Y.Q. analytical approaches. Y.L. and R.X. helped resource estimation. J.Y. and X.Y. contributed computations and simulations. D.Z. and D.T. led the writing with input from S.J.D., Y.L., and Q.Z.

## Competing interests

The authors declare no competing interests.
