## [Peer Review File · Nature Communications]

REVIEWER COMMENTS

Reviewer #1 (Remarks to the Author):

Review of NCOMMS-23-26021: Increases in extreme power shortage events of wind-solar supply systems worldwide

The study is important in addressing the extreme power shortage risk supported mainly by wind and solar, which would help inform the renewable policy in the future. However, the study still requires a considerable amount of work or revision before it fulfills the standard requirements of the journal.

Main Comments:

1. The integrated and solar thermal technology are not in consideration for the solar supply, which challenges the validity and soundness.
2. The wind speed varies dramatically even on a minute-scale, which means that a higher temporal resolution is required.
3. A fixed wind-solar share (59.05%) was used to calculate the power shortage (the gap between wind-solar supply and electricity demand), which means that the influence of the electricity grid was neglected. The reasonableness should be addressed more in discussion or other sections.
4. Solar and wind technology has made great progress since 2000. For example, the capacity proportion of the dual-axis tracking type has been increasing surprisingly, which means that the assumption of dual-axis tracking type would underestimate the actual reliability of 1980-2000.
5. This study has incorporated actual solar and wind capacity evolution/government planning trajectory as the constraints, right? It is difficult to understand the reason for estimating the installed capacity by electricity demand (Line 376-377) while neglecting economics, geographics, and renewable policies in the model.
6. The study addressed that the low wind speed is the main driver (rather than the radiation) for the power shortage risk, which should be addressed more in the Abstract (as an important highlight).
7. The long-duration and low-reliability risks were selected to describe power shortage events, but which one will damage the future energy systems much more? What is the combined risk of long-duration and low-reliability? The discussion on those problems would complete the study better.
8. The results were derived based on the comparison between 1980-2000 and 2001-2022. It would be better to polish the trends by splitting the periods and comparing them over 4 decades (1980-1990, 1991-2000, 2001-2010, 2011-2022).

9. Other Questions on the manuscript

☐ (1)The introduction has analyzed the research gap, but the challenge for the study is not so evident.

☒ (2) Figure 1 has only mapped the case of $P < 0.001$, while Figure 3 only mapped the case of $P < 0.05$. why? More details on the “P” are necessary.

☒ (3) It would be better to name the major economies in Figure 4, like China and UK.

☒ (4) It is necessary to improve the figures (for example, the texts were hidden by the figure body as in Extended Data Fig. 1 and Extended Data Fig. 2).

Reviewer #2 (Remarks to the Author):

The manuscript investigates extreme power shortage events in wind-solar systems, utilizing 43 years of reanalysis climatological data from MERRA-2 across 42 countries. The authors aim to understand the dynamics and implications of these events on a national and global scale, drawing insights from these statistical analyses.

Major Comments:

1. The manuscript's analysis appears to lack a robust design that sufficiently supports the conclusions drawn. The utilization of a single reanalysis product (MERRA-2) without a clear rationale or comparison with other available reanalysis datasets undermines the comprehensiveness and reliability of the findings. A more detailed justification for the exclusive selection of MERRA-2 or an incorporation of multiple reanalysis products, including ERA series, would strengthen the study's credibility. 2. Similarly, the choice of 42 'major' (according to the authors) countries for analysis is not adequately justified. The criteria for selecting these specific countries, and their representativeness, are not clearly articulated. A more transparent criterion for country selection, possibly based on regional representation or other relevant factors, is essential to validate the study's conclusions at national, regional, and global levels.

A refined methodological approach, considering additional reanalysis products and a broader or more systematic country selection, is recommended to bolster the study's credibility.

While I have several minor comments, I believe addressing the fundamental issues related to research design and methodology first is paramount.

Reviewers Comments:

Reviewer #1 (Remarks to the Author):

The study is important in addressing the extreme power shortage risk supported mainly by wind and solar, which would help inform the renewable policy in the future. However, the study still requires a considerable amount of work or revision before it fulfills the standard requirements of the journal.

Response: We thank the Referee for the positive and accurate summary of our work and for the fair and thoughtful comments below. We have made a number of revisions in response, and believe the manuscript has been substantially improved. A point-to-point response is presented below.

1. The integrated and solar thermal technology are not in consideration for the solar supply, which challenges the validity and soundness.

Response: We thank the Referee for the constructive comment. We totally agree that concentrating solar power (CSP) technology has occupied a crucial position in solar power system, due to its combination with energy storage system that enables solar power generation more stable (Pfenninger et al., 2014; Kennedy et al., 2022). Our manuscript focuses mainly on the impact of climate change on the reliability of wind-solar supply systems through excluding flexible electricity resources such as energy storage to address the potential power supply gaps, and thus CSP technology is not in consideration for the solar supply.

Given potential importance of CSP in the reliability of solar supply system, we have added a sensitivity test that integrates CSP and photovoltaic solar technologies based on their future supply shares by this mid-century. The future shares of CSP and photovoltaic solar technologies are collected from the International Institute for Applied Systems Analysis (IIASA) (<https://data.ece.iiasa.ac.at/ar6/>) (Supplementary Table 5). The results with consideration of CSP technologies do not alter our main conclusions (Supplementary Figs. 25–29).

As shown in Supplementary Figs. 25 and 26, our estimates combined with CSP technology also shows an uptrend trend in long-duration extreme power shortage events since 1980, and such an increasing trend in long-duration events is of global significance. We also examined the change in low-reliability events during the period of 1980–2022, and the results can also capture the increasing low-reliability events across the world, despite some different estimates in Middle East compared to our initial results (see Figure 3, Supplementary Figs. 27 and 28). Besides, we further investigate the relationship between changes in extreme events and variabilities in climatic variables, which still supports our finding that uptrend in extremely low wind speed is behind the growing extreme power shortage events (see Supplementary Fig. 29). We have added the sensitivity test in revised Discussion (Lines 258–261) and Supplementary Text 7.

Discussion

Besides, CSP, as a potential important technology towards the reliability of solar supply system³⁸, is also considered into our technology sensitivity tests according its future supply share (Supplementary Table 5), which shows the consistent findings (see supplementary Text 7 and Supplementary Figs. 25–29).

Supplementary Text 7. Sensitive tests of power shortage events with consideration of concentrating solar power technology (CSP)

Our study mainly involves the impact of climate change on the reliability of wind-solar supply systems through excluding flexible electricity resources such as energy storage, and thus CSP technology is not in consideration for the solar supply. However, given potential importance of CSP technology in the reliability of solar supply system, we have added a sensitivity test that integrates CSP and photovoltaic solar technologies based on their future supply shares by the mid-century. The future shares of CSP and photovoltaic solar technologies are obtained from the International Institute for Applied Systems Analysis (IIASA) (<https://data.ece.iiasa.ac.at/ar6/>; Supplementary Table 5). We calculated CSP supply output potential according to solar direct irradiance and ambient temperature^{7,14}.

The results with consideration of CSP technologies do not alter our main conclusions (see Supplementary Figs. 25–29). As shown in Supplementary Figs. 25 and 26, our estimates combined with CSP technology also shows an uptrend trend in long-duration extreme power shortage events since 1980, and such an increasing trend is of global significance. We also examined the change in low-reliability events during the period of 1980–2022, and the results can also capture the increasing low-reliability events across the world, despite some different estimates in Middle East compared to our initial results (see Figure 3, Supplementary Figs. 27 and 28). Besides, we further investigate the relationship between changes in extreme events and variabilities in climatic variables, which still supports our finding that uptrend in extremely low wind speed is behind the growing extreme power shortage events (Supplementary Fig. 29).

Supplementary Figure 25 | Interannual variability of extreme long-duration events of wind-solar supply system with consideration of CSP technology. a-c, Interannual changes in the frequency (a), duration (b), and intensity (c) of extreme long-duration events since 1980. The coefficients at the top of the panels indicate robust Theil-Sen's slopes and their corresponding P values, examined by Mann-Kendall (MK) test. The right boxplots denote the difference before 2000 (blue) and after 2000 (red) examined by analysis of variance (ANOVA). *** represents the significance under the level of $P < 0.001$. Black dashed lines denote linear fitting of annual average values across 178 countries. d-f, Rank ordering of the annual average frequency (d), duration (e), and intensity (f) of extreme long-duration events across the surveyed 43 years, in which years between 1980 and 2000 are labeled with black and those between 2001 and 2022 are labeled with red.

Supplementary Figure 26 | Maps of extreme long-duration events with CSP technology during the period 1980–2000 and 2001–2022. a-c, Annual average frequency (a), duration (b), and intensity (c) of extreme long-duration events during the period 1980–2000. **d-f,** Annual average frequency (d), duration (e), and intensity (f) of extreme long-duration events during the period 2001–2022. **g-i,** Changes in annual average frequency (g), duration (h), and intensity (i) of extreme long-duration events between 1980–2000 and 2001–2022.

Supplementary Figure 27 | Changes in extreme low-reliability events with CSP technology since 1980. a-c, Changes in annual average frequency (a), duration (b), and intensity (c) of extreme low-reliability events between 1980–2000 and 2001–2022. The left-bottom boxplots denote the difference between 1980–2000 (blue) and 2001–2022 (red) examined by ANOVA, where ** and *** represent

the significance under the level of $P < 0.01$ and $P < 0.001$, respectively. **d**, Changes in extreme low-reliability events for 42 major countries between 1980–2000 and 2001–2022.

Supplementary Figure 28 | Relationship of changes in extreme long-duration and low-reliability events with consideration of CSP technology. Changes in frequency (**a**), duration (**b**), and intensity (**c**) of extreme long-duration and low-reliability events for the surveyed 178 countries between 1980–2000 and 2001–2022. Dashed red lines with shallow red shading are linear fittings between the two types of events and their 95% confidence intervals.

Supplementary Figure 29 | Changes in extreme power shortage events with extremely low wind speed and solar radiation with consideration of CSP. a-c, Relationship between anomalies in frequency (a), duration (b), and intensity (c) of extreme power shortage events and anomalies in annual hours of extremely low solar radiation and wind speed at hub height for individual countries. **d,** Relationship between change in annual average wind speed and change in annual hours of extremely low wind speed between 1980–2000 and 2001–2022. **e,** Relationship between the relative change in annual hours of extremely low wind speed and the relative change in extreme power shortage events between 1980–2000 and 2001–2022. The relative change above 100% is visualized as 100% change. The lines and circles denote individual country’s change and the average changes across 178 surveyed countries, respectively. The surrounding digits represent the order number of the 178 countries listed in Supplementary Table 6.

Supplementary Table 5 | Predicted shares of CSP and PV to the total solar supply by mid-century

Geographical zones	CSP share	PV share
North America	8.45%	91.55%
United States	1.94%	98.06%
Mexico	6.64%	93.36%
Central and South America	5.73%	94.27%
Brazil	11.08%	88.92%
Argentina	6.70%	93.30%
Chile	0.06%	99.94%
Venezuela	0.30%	99.70%
Europe	1.16%	98.84%
Germany	0.79%	99.21%
France	0.69%	99.31%
UK	0.95%	99.05%
Russia	2.46%	97.54%
Austria	0.00%	100.00%
Belgium	0.00%	100.00%
Poland	0.00%	100.00%
Denmark	0.00%	100.00%
Finland	0.00%	100.00%
Greece	1.50%	98.50%
Turkey	2.06%	97.94%
Africa	8.45%	91.55%
Morocco	0.05%	99.95%
Angola	0.03%	99.97%
Algeria	0.01%	99.99%
Egypt	0.01%	99.99%
Kenya	0.04%	99.96%
South Africa	19.10%	80.90%
Middle East	15.64%	84.36%
Saudi Arabia	22.90%	77.10%
Asia	3.22%	96.78%
China	1.21%	98.79%

India	5.63%	94.37%
Japan	0.14%	99.86%
Indonesia	12.13%	87.87%
Pakistan	14.98%	85.02%
Oceanic	9.89%	90.11%
Australia	2.87%	97.13%

References:

Pfenninger, S. et al. Potential for concentrating solar power to provide baseload and dispatchable power. *Nat. Clim. Chang.* 4, 689–692 (2014).

Kennedy, K. M. et al. The role of concentrated solar power with thermal energy storage in least-cost highly reliable electricity systems fully powered by variable renewable energy. *Adv. Appl. Energy* 6, 100091 (2022).

2. The wind speed varies dramatically even on a minute-scale, which means that a higher temporal resolution is required.

Response: We thank the Referee for the constructive comment. We totally agree that minute-scale dramatical change of wind speed may heavily affect the estimate of wind power output. It is unfortunate that minute-scale wind speed and power demand data remain unavailable, which limited our modeling and simulation of extreme power shortage events at a higher temporal resolution. Therefore, the point of developing and opening minute-scale meteorology data to support follow-up refined analysis on extreme power shortage events has been discussed in revised Discussion section (Lines 243–247) and Supplementary Text 5 as suggested.

Discussion

Our estimates are restricted by several potential limitations and uncertainties like meteorological data and technology assumption. In terms of meteorological data, on one hand, current hour-scale resolution may not capture the dramatical change in wind speed at the minute scale. It limits our refined knowledge on extreme power shortage events, which in turn calls for future access to high-resolution meteorological data (Supplementary Text 5).

Supplementary Text 5. Expanded discussion on temporal resolution of wind speed data

Although this study that evaluates the reliability of wind-solar power supply systems based upon an hourly climatologic reanalysis data can capture the variability characteristics of potential power shortage events, wind speed varies dramatically even on a minute-scale. Therefore, it is necessary to perform a higher temporal resolution of analyses. However, it is unfortunate that minute-scale wind speed and power demand data remain unavailable, which limited our modelling and simulation of extreme power shortage events at a higher temporal resolution analysis. Thus, minute-scale meteorology data should support follow-up refined analysis on extreme power shortage events, if minute-scale data become accessible in the future.

3. A fixed wind-solar share (59.05%) was used to calculate the power shortage (the gap between wind-solar supply and electricity demand), which means that the influence of the electricity grid was neglected. The reasonableness should be addressed more in discussion or other sections.

Response: We thank the Referee for this important comment, and we are sorry for confusing the Referee on the wind-solar share applied for this work. Here, we employed regional and national projections of wind-solar share by mid-century from International Energy Agency (IEA, see Supplementary Table 2), and the global average of wind-solar share is 59.05%. We have clarified the associated description texts concerning to wind-solar share in Introduction (Lines 63–66), Discussion (Lines 262–267), and Method (Lines 403–407) sections as suggested.

Introduction

Note that we do not assume a fixed wind-solar share for all the countries, but rather different regional and national wind-solar shares are applied to estimate potential power shortage events for each country, according to the International Energy Agency (IEA; Supplementary Table 2)²⁵.

Discussion

Our study may also be affected by wind-solar supply share and demand profile. To be specific, we assume an electricity system with a certain share of wind-solar energy by mid-century around the world (Supplementary Table 2)²⁵; however, the future share of wind-solar power likely varies and is even more aggressive for individual countries³⁹. Our additional sensitivity tests, which assumed 80% and 100% wind-solar hybrid systems globally, even exhibited a greater intensity of potential extreme power shortage events (Supplementary Figs. 30 and 31).

Method

Share_{sw} represents the predicted share of wind and solar to the total electricity generation by mid-century according to IEA (Supplementary Table 2)²⁵. The IEA predicts future wind-solar share of regions and major countries (e.g., United States, India, and China), yet the shares of wind-solar power in small countries are unavailable. We therefore assigned regional wind-solar shares to countries without prediction.

Supplementary Table 2 | Share of wind and solar to the total electricity generation in 2050 based upon IEA

Regions	Share
World	59.05%
North America	67.45%
United States	72.62%
Central and South America	56.36%
Brazil	54.15%
Europe	63.13%
European Union	67.47%

Africa	56.80%
Middle East	53.67%
Eurasia	13.40%
Russia	7.48%
Asia Pacific	59.41%
China	60.75%
India	71.63%
Japan	38.68%
Southeast Asia	42.49%

4. Solar and wind technology has made great progress since 2000. For example, the capacity proportion of the dual-axis tracking type has been increasing surprisingly, which means that the assumption of dual-axis tracking type would underestimate the actual reliability of 1980-2000.

Response: We thank the Referee for this constructive comment. We totally agree that achieved solar and wind technology progress could largely increase the power output reliability, especially in recent decades (IPCC, 2022; IRENA, 2022). The same tracking type applied across all study period (i.e., 1980–2022) is to separate the impact of climate change (only varying the meteorological conditions across the whole period). Therefore, to clarify the scientific issue focused by this work, we have accordingly changed our title as “*Climate change impacts on the power shortage events of wind-solar supply systems worldwide*”.

Meanwhile, we have added new simulations of single-axis tracking type during the period of 1980–2000. It shows that the reliability of wind-solar supply systems would be slightly underestimated across the world, with the change in system reliability ranging from -0.008 to 0.002 (see Supplementary Fig. 20). We further examined the impact of solar tracking types on extreme power shortage events of wind-solar supply systems during the period through 1980–2000. The results based upon single-axis tracking system likely allow an underestimate of extreme events in middle- and low-latitude countries but an overestimate of extreme events in high-latitude countries (see Supplementary Fig. 21).

Additionally, we investigate the influence of tracking system progress on trends in extreme power shortage events since 1980, during which we assume one-axis solar tracking system over the period of 1980–2000 but dual-axis solar tracking system between 2001 and 2022. We find that, solar tracking type progress does not alter the observed increasing trends in both long-duration and low reliability events of wind-solar electricity system since 1980s (Supplementary Figs. 22–24). We have added these sensitivity tests of single-axis tracking type and tracking system progress in Discussion section (Lines 252–258) and Supplementary Text 6 in revised manuscript.

Discussion

As for technology assumption when estimating the solar capacity factors, our analyses based upon dual-axis tracking system may result in potential uncertainty of system reliability and power shortage events, due to the continued progress of solar tracking systems. Our tests with consideration of single-axis tracking type and tracking system progress show lower system

reliabilities but similar increasing trends in both long-duration and low reliability events of wind-solar supply systems (Supplementary Text 6 and Supplementary Figs. 20–24).

Supplementary Text 6. Sensitive tests of system reliability and power shortage events with consideration of single-axis tracking system and solar tracking technology progress

Our study based upon dual-axis tracking system may result in potential uncertainty of system reliability and power shortage events. Therefore, we have added new simulations of single-axis tracking type during the period of 1980–2000. It shows that the reliability of wind-solar supply systems would be slightly underestimated across the world, with the change in system reliability ranging from -0.008 to 0.002 (see Supplementary Fig. 20). We further examined the impact of solar tracking types on extreme power shortage events of wind-solar supply systems during the period through 1980–2000. The results based upon single-axis tracking system result likely in an underestimate of extreme events in middle- and low-latitude countries but an overestimate of extreme events in high-latitude countries (see Supplementary Fig. 21). Additionally, we investigate the influence of tracking system progress on the trend in extreme power shortage events since 1980, during which we assume one-axis solar tracking system over the period of 1980–2000 but dual-axis solar tracking system between 2001 and 2022. We find, that solar tracking type progress does not alter the observed increasing trends in both long-duration and low reliability events of wind-solar electricity system since 1980s (Supplementary Figs. 22–24).

Supplementary Figure 20 | The impacts of solar tracking systems on system reliability across the world. The changes in annual average reliability between single-axis and dual-axis solar tracking systems during the period of 1980 and 2000.

Supplementary Figure 21 | The impacts of solar tracking systems on extreme power shortage events across the world. a-c. The changes in frequency (a), duration (b), intensity (b) of long-duration events between single-axis and dura-axis solar tracking systems during the period of 1980 and 2000. **d-f.** The changes in frequency (d), duration (e), intensity (f) of low-reliability events between single-axis and dura-axis solar tracking systems during the period of 1980 and 2000.

Supplementary Figure 22 | Interannual variability of extreme long-duration events with consideration of solar tracking system progress. **a-c**, Interannual changes in the frequency (**a**), duration (**b**), and intensity (**c**) of extreme long-duration events since 1980. The coefficients at the top of the panels indicate robust Theil-Sen's slopes and their corresponding P values, examined by Mann-Kendall (MK) test. The right boxplots denote the difference before 2000 (blue) and after 2000 (red) examined by analysis of variance (ANOVA). *** represents the significance under the level of $P < 0.001$. Black dashed lines denote linear fitting of annual average values across 178 countries. **d-f**, Rank ordering of the annual average frequency (**d**), duration (**e**), and intensity (**f**) of extreme long-duration events across the surveyed 43 years, in which years between 1980 and 2000 are labeled with black and those between 2001 and 2022 are labeled with red.

Supplementary Figure 23 | Maps of extreme long-duration events with consideration of solar tracking system progress. **a-c**, Annual average frequency (**a**), duration (**b**), and intensity (**c**) of extreme long-duration events during the period 1980–2000. **d-f**, Annual average frequency (**d**), duration (**e**), and intensity (**f**) of extreme long-duration events during the period 2001–2022. **g-i**, Changes in annual average frequency (**g**), duration (**h**), and intensity (**i**) of extreme long-duration events between 1980–2000 and 2001–2022.

Supplementary Figure 24 | Changes in extreme low-reliability events with consideration of solar tracking system progress. a-c, Changes in annual average frequency (a), duration (b), and intensity (c) of extreme low-reliability events between 1980–2000 and 2001–2022. The left-bottom boxplots denote the difference between 1980–2000 (blue) and 2000–2022 (red) examined by ANOVA, where ** and *** represent the significance under the level of $P < 0.01$ and $P < 0.001$, respectively. **d,** Changes in extreme low-reliability events for 42 major countries between 1980–2000 and 2001–2022.

References:

IPCC. 2022. Climate Change 2022: Mitigation of Climate Change. doi: 10.1007/978-0-85729-244-5_1.

IRENA. 2022. Renewable Technology Innovation Indicators: Mapping Progress in Costs, Patents and Standards.

5. This study has incorporated actual solar and wind capacity evolution/government planning trajectory as the constraints, right? It is difficult to understand the reason for estimating the installed capacity by electricity demand (Line 376-377) while neglecting economics, geographics, and renewable policies in the model.

Response: We appreciate the Referee for the meaningful comment. We totally agree that the development of solar and wind capacity depends on the comprehensive factors such as economics, geographics, and renewable policies. Here we estimated the installed capacity by electricity demand in this study for the following two reasons.

First, the actual solar and wind capacity evolution/government planning trajectory is not incorporated as the year-based constraints. As mentioned in the above comment, single regional and national projections of wind-solar share by mid-century are used for separating the impact of climate change (only varying the meteorological conditions across this period), which aims to identify and highlight the increasing risks of high-penetration wind and solar power supply system under the fact of climate change. Therefore, to clarify the scientific issue focused by this work, we have changed our title as “*Climate change impacts on the power shortage events of wind-solar supply systems worldwide*”.

Second, wind-solar shares from the IEA projection, that considers the latest energy market, economic and demographic forces, as well as announced climate-related commitments by governments and non-governmental organizations (IEA, 2022), are also used for the estimate of installed capacity. To be specific, the derived installed capacity enables that the wind and solar power generation can satisfy a certain share of annual total power demand. Then, the resulting installed capacity, together with hour-scale capacity factor is used to simulate hourly wind-solar output and potential extreme power events. Moreover, this method based upon hourly capacity factors also takes account of potential geographical constraints such as country’s land areas and latitudes, which have been confirmed to affect wind-solar ratio, install capacities, and system reliability (Liu et al., 2020; Tong et al., 2021). In order to avoid this confusion, we have revised the description texts of installed capacity calculation in *Method* section of revised manuscript as suggested.

Additionally, as the potential importance of economics and renewable policies towards the wind-solar share of power generation, we have further performed a series of sensitivity tests based upon REMIND model. REMIND is a well-known integrated assessment model (IAM) that coupled macroeconomic growth and renewable polices into the assessment of future electricity supply mix (e.g., wind and solar) (Luderer et al., 2022; Ueckerdt et al., 2017). Here, we employed the wind-solar share in 2050 under 1.5 °C climate target projected by the REMIND model (Supplementary Table 4), to calculate the wind-solar installed capacities and extreme power shortage events. As shown in supplementary Figs 32–36, the results also underpin growing extreme power shortage events of wind-solar electricity system across the world, albeit with decline in extreme events in a few countries such as Sudan and Saudi Arabia (see Supplementary Figs. 32–35). Furthermore, we explored the potential mechanisms underlying such phenomenon, which also highlights extremely low wind speed may be responsible for the observed growing extreme power shortage events (see Supplementary Fig. 36). These sensitivity tests of economics and renewable policies based upon REMIND model have been discussed in Discussion section (Lines 268–272) and Supplementary Text 8 of the revised manuscript.

Discussion

Moreover, another sensitivity test of wind-solar share in 2050 given potential importance of economics and renewable policies (1.5 °C climate target from the REMIND model^{27,40}) was performed, and the results do not change the robustness of our main findings concerning to

growing extreme power shortage events and its driving forces (Supplementary Text 8 and Supplementary Figs. 32–36).

Method

Installed capacities and hourly supply of wind and solar powers.

The installed capacity of wind and solar for individual countries are calculated based upon electricity demand and the future projected share of wind and solar⁴:

$$Capacity_w = Dem_{avg} \times Share_{sw} \times Ratio_w / CF_{wm} \quad (9)$$

$$Capacity_s = Dem_{avg} \times Share_{sw} \times (1 - Ratio_w) / CF_{sm} \quad (10)$$

Where, $Capacity_w$ and $Capacity_s$ denote the installed capacities for wind and solar, respectively. DEM_{avg} is average hourly electricity demand. $Share_{sw}$ represents the predicted share of wind and solar to the total electricity generation by mid-century according to IEA (Supplementary Table 2)²⁵. The IEA predicts future wind-solar share of regions and major countries (e.g., United States, India, and China), yet the shares of wind-solar power in small countries are unavailable. We therefore assigned regional wind-solar shares to countries without prediction. $Ratio_w$ and $1 - Ratio_w$ are the optimal ratio of wind and solar energy to wind-solar systems (Supplementary Fig. 39), respectively, which is described in Supplementary Text 9 in detail. CF_{wm} and CF_{sm} indicate area-weighted average wind and solar capacity factors during the study period through 1980–2022, respectively.

The resulting installed capacities of wind and solar powers are further applied to calculate hourly wind and solar supply by multiplying hourly wind and solar capacity factor:

$$Sup_{w,t} = Capacity_w \times CF_{w,t} \quad (11)$$

$$Sup_{s,t} = Capacity_s \times CF_{s,t} \quad (12)$$

Where, $Sup_{w,t}$ and $Sup_{s,t}$ represent wind and solar electricity supply at hour t , respectively. $CF_{w,t}$ and $CF_{s,t}$ refer to the area-weighted average capacity factors of wind and solar power at hour t , respectively.

Supplementary Text 8. Sensitive tests of economics and renewable policy through using the wind-solar share predicted by REMIND model.

Our study estimates the install capacities of wind and solar power using the IEA predicted wind-solar share that have considered driving factors including economics, geographics, and renewable policies. However, as the potential importance and uncertainty of economics and renewable policies towards the wind-solar share of power generation, we have further performed a series of sensitivity tests based upon REMIND model^{14,15}. REMIND is a well-known integrated assessment model (IAM) that coupled macroeconomic growth and renewable

polices into the assessment of future electricity supply mix (e.g., wind and solar). Here, we employed the wind-solar share in 2050 under 1.5 °C climate target projected by the REMIND model (Supplementary Table 4), to calculate the wind-solar installed capacities and the extreme power shortage events. As shown by Supplementary Figs. 32–36, the results also underpin growing extreme power shortage events of wind-solar electricity system across the world, albeit with downtrends in a few countries such as Sudan and Saudi Arabia (see Supplementary Figs. 33–35). Furthermore, we explored the potential mechanisms underlying such phenomenon, which also highlights extremely low wind speed may be responsible for the observed growing extreme power shortage events (see supplementary Fig. 36).

Supplementary Figure 32 | Interannual variability in extreme long-duration events of wind-solar supply system using the wind-solar shares predicted by REMIND model. a-c, Interannual changes in the frequency (a), duration (b), and intensity (c) of extreme long-duration events since 1980. The coefficients at the top of the panels indicate robust Theil-Sen’s slopes and their corresponding P values, examined by Mann-Kendall (MK) test. The right boxplots denote the difference before 2000 (blue) and after 2000 (red) examined by analysis of variance (ANOVA). *** represents the significance under the level of $P < 0.001$. Black dashed lines denote linear fitting of annual average values across 178 countries. d-f, Rank ordering of the annual average frequency (d), duration (e), and intensity (f) of extreme long-duration events across the surveyed 43 years, in which years between 1980 and 2000 are labeled with black and those between 2001 and 2022 are labeled with red.

Supplementary Figure 33 | Maps of extreme long-duration events based upon the wind-solar shares predicted by REMIND model. a-c, Annual average frequency (a), duration (b), and intensity (c) of extreme long-duration events during the period 1980–2000. **d-f,** Annual average frequency (d), duration (e), and intensity (f) of extreme long-duration events during the period 2001–2022. **g-i,** Changes in annual average frequency (g), duration (h), and intensity (i) of extreme long-duration events between 1980–2000 and 2001–2022.

Supplementary Figure 34 | Changes in extreme low-reliability events calculated through the wind-solar shares predicted by REMIND model. a-c, Changes in annual average frequency (a), duration (b), and intensity (c) of extreme low-reliability events between 1980–2000 and 2001–2022. The left-bottom boxplots denote the difference between 1980–2000 (blue) and 2000–2022 (red) examined by ANOVA,

where *** represent the significance under the level of $P < 0.001$. **d**, Changes in extreme low-reliability events for 42 major countries between 1980–2000 and 2001–2022.

Supplementary Figure 35 | Relationship of changes in extreme long-duration and low-reliability events using the wind-solar shares predicted by REMIND model. Changes in frequency (a), duration (b), and intensity (c) of extreme long-duration and low-reliability events for the surveyed 178 countries between 1980–2000 and 2001–2022. Dashed red lines with shallow red shading are linear fittings between the two types of and their 95% confidence intervals. VEN and D. Congo denote Venezuela and Democratic Republic of the Congo, respectively.

Supplementary Figure 36 | Changes in extreme power shortage events with extremely low wind speed and solar radiation using the wind-solar shares in REMIND model. a-c, Relationship between anomalies in frequency (a), duration (b), and intensity (c) of extreme power shortage events and anomalies in annual hours of extremely low solar radiation and wind speed at hub height for individual countries. **d,** Relationship between change in annual average wind speed and change in annual hours of extremely low wind speed between 1980–2000 and 2001–2022. **e,** Relationship between the relative change in annual hours of extremely low wind speed and the relative change in extreme power shortage events between 1980–2000 and 2001–2022. MOZ denotes Mozambique. The relative change above 100% is visualized as 100% change. The lines and circles denote individual country’s change and the average changes across 178 surveyed countries, respectively. The surrounding digits represent the order number of the 178 countries that are listed in Supplementary Table 6.

Supplementary Table 4 | Share of wind and solar to the total electricity generation predicted by REMIND model^{15,18}

Geographical Zones	Share of wind and solar
Canada, Australia, New Zealand	63.93%
China	67.51%
Europe	87.62%
India	90.67%
Japan	64.70%
Latin America	73.37%
Middle East	91.93%
Non-EU Europe	73.16%
Other Asia	87.77%
Reforming countries	76.48%
Sub-Saharan Africa	84.13%
USA	90.62%
World	82.95%

References:

IEA. World Energy Outlook 2022. International Energy Agency (IEA) (2022).

Liu, L. et al. Optimizing wind/solar combinations at finer scales to mitigate renewable energy variability in China. *Renew. Sustain. Energy Rev.* 132, 110151 (2020).

Luderer, G. et al. Impact of declining renewable energy costs on electrification in low-emission scenarios. *Nat. Energy* 7, (2022).

Ueckerdt, F. et al. Decarbonizing global power supply under region-specific consideration of challenges and options of integrating variable renewables in the REMIND model. *Energy Econ.* 64, 665–684 (2017).

Tong, D. et al. Geophysical constraints on the reliability of solar and wind power worldwide. *Nat. Commun.* 12, 6146 (2021).

6. The study addressed that the low wind speed is the main driver (rather than the radiation) for the power shortage risk, which should be addressed more in the Abstract (as an important highlight).

Response: Thanks for this valuable suggestion. We have rephrased the Abstract (Lines 12–14) to highlight that extremely low wind speed is the main driver instead of solar radiation as suggested.

Abstract

Instead of extremely low solar radiation, this observed uptrends in extreme power shortage events are primarily driven by increases in extremely low wind speeds, which however are strongly disproportionated.

7. The long-duration and low-reliability risks were selected to describe power shortage events, but which one will damage the future energy systems much more? What is the combined risk of long-duration and low-reliability? The discussion on those problems would complete the study better.

Response: Good points. Towards the security of future electricity systems, long-duration and low-reliability power shortage events would have unequal risks and may call for different scales and types of flexible energy to help smooth fluctuations, whose challenges in turn likely depend on the future innovative storage technology development and the scale of stable backup fossil-fuel-based power capacity (Pan et al., 2021; Rinaldi et al., 2021; IEA 2022). For example, if cost-effective long-duration storage technology matures with limited backup fossil capacity in the future, the threat of low-reliability events probably outweighs that of long-duration events. On the contrary, long-duration events may be a severer threat to power systems than low-reliability events.

The combination of the two types defined power shortage events probably bring much more serious risks than either type of event. In general, long-duration events tend to trigger electricity outage accidents which last for a long period, while low-reliability events likely lead to short but large-scale supply failure. Unfortunately, our results demonstrated that both long-duration events and low-reliability events show an increasing trend, suggesting that future electricity system probably experience both longer and higher power gaps in the context of climate change. Such a combined risk may call for a more urgent need to backup electricity generation, and thus strategic integration of multiple flexibility resources would be a feasible solution to address such combined problems in the future.

We have extended the Discussion (Lines 285–300) in revised manuscript as suggested.

Discussion

Towards the security of future electricity systems, long-duration and low-reliability power shortage events would have unequal risks and may call for different scales and types of flexible energy to help smooth fluctuations, whose challenges in turn likely depend on the future storage technology development and the scale of stable backup fossil-fuel-based power capacity^{19,20,25}. For example, if cost-effective long-duration storage technology matures with limited backup fossil capacity kept in the future, the threat of low-reliability events probably outweighs that of long-duration ones. On the contrary, long-duration events may be a severer threat to power systems than low-reliability ones. On top of that the combination of these two types defined events probably bring much more serious risks than either type of event. Long-duration events tend to trigger electricity outage accidents which last for a long period, while low-reliability events likely lead to short but large-scale supply failure. Unfortunately, our results demonstrated that both show an increasing trend, suggesting that future electricity system probably experience both longer and higher power gaps in the context of lasting climate change.

The combined risk may call for a more urgent need to backup electricity generation, and thus strategic integration of multiple flexibility resources would be a feasible solution to address such electricity security problems in the future.

References:

Pan, G. et al. Assessment of plum rain's impact on power system emissions in Yangtze-Huaihe River basin of China. *Nat. Commun.* **12**, 6156 (2021).

Rinaldi, K. Z., Dowling, J. A., Ruggles, T. H., Caldeira, K. & Lewis, N. S. Wind and solar resource droughts in California highlight the benefits of long-term storage and integration with the western interconnect. *Environ. Sci. Technol.* **55**, 6214–6226 (2021).

IEA. *World Energy Outlook 2022*. International Energy Agency (IEA) (2022).

8. The results were derived based on the comparison between 1980–2000 and 2001–2022. It would be better to polish the trends by splitting the periods and comparing them over 4 decades (1980–1990, 1991–2000, 2001–2010, 2011–2022).

Response: We thank the Referee for this valuable comment. We have added additional analyses in potential power shortage events over the periods of 1980–1990, 1991–2000, 2001–2010, and 2011–2022 (Supplementary Figs. 7–9). The resulting interdecadal analyses indicate that power shortage events have largely increased since 1980s (particularly since 1990s), but extreme power shortage events represented a much more obvious growth between 1980–2000 and 2001–2022 (Figs. 1–3).

We have added such interdecadal summary in Results (Lines 96–105, 114–117, and 142–145) since the consistent conclusions of 20-year and 10-year split, and plenty of analysis in Supplementary Text 2 of revised manuscript, but we can also place them in the main text if necessary.

Results

Variability in extreme long-duration shortage events

However, extreme long-duration events showed much more complicated decadal changes, with no significant disparities between 1980–1990 and 1991–2000 ($P < 0.05$), but obvious uptrends regardless of its frequency, duration, and intensity between 1991–2000 and 2001–2010 (Supplementary Text 2 and Supplementary Figs. 7–9). We further ranked the annual average frequency, duration, and intensity of extreme long-duration events over the past 43 years (Figs. 1d–f). Our results revealed that the greatest frequency, duration, and intensity of extreme long-duration events occurred intensively in the period between 2001 and 2022, especially the period between 2011 and 2022. For instance, ~82% of periods from 2001 to 2022 have been top 20 frequent years of extreme long-duration events (Fig. 1d and Supplementary Fig.7).

Such spatial discrepancy was even more apparent on a decadal scale. For example, duration of extreme shortage events represented the highest increase by +106 hours in Chad between 2001–2010 and 2011–2022, but the largest decrease by in Seychelles between 1991–2000 and 2001–2010 (Supplementary Fig. 8).

Change in extreme low-reliability shortage events

This spatial disparity of variability in frequency of extreme long-duration events would be even larger on a decadal scale, with ranging from nearly +13 in Benin between 1991–2000 and 2001–2010 to only around –12 in Suriname between 2001–2010 and 2011–2022 (Supplementary Fig. 9).

Supplementary Text 2. Decadal analyses for extreme power shortage events of wind-solar supply systems

The observed extreme long-duration events showed much more complicated decadal changes, with no significant disparities between 1980–1990 and 1991–2000 ($P < 0.05$), but obvious uptrends regardless of its frequency, duration, and intensity between 1991–2000 and 2001–2010 (Supplementary Fig. 7–9). We further ranked the annual average frequency, duration, and intensity of extreme long-duration events over the past 43 years (Supplementary Fig. 7d–f). Our results revealed that the greatest frequency, duration, and intensity of extreme long-duration events occurred intensively in the period of 2001–2022 (the period of 2011–2022 in particular). For instance, the period from 2001 to 2022 accounted for 90% of the top 20 frequent years of extreme long-duration events, and the period of 2011–2022 occupied 60% (Supplementary Fig. 7d). This case is even more apparent for the top 10 frequent years, all of which emerged over the period after 2000 and 70% of which occurred during the period after 2011 (Supplementary Fig. 7d).

We further illustrated that the spatial pattern of the observed change in extreme long-duration events since 1980s (Supplementary Fig. 8). The number of countries with increasing extreme long-duration events is usually higher than that of countries with decreasing extreme long-duration events. For example, there are 64% countries with uptrends in frequency of extreme long-duration events during the periods between 1990 and 2000. However, such upward trends in extreme long-duration events at the global scale mask large spatial disparities across various countries across the past four decades. For example, duration of extreme shortage events represented the highest increase by +106 hours in Chad between 2001–2010 and 2011–2022, but the largest decrease by in Seychelles between 1991–2000 and 2001–2010 (Supplementary Fig. 8).

Apart from extreme long-duration events during, extreme low-reliability events are also severe challenges for maintaining wind-solar system stability. There are 168 out of 178 countries where extreme low-reliability events have emerged since 1980, except for countries

with comparatively large land areas, such as Russia and China, as a result of their strong spatial aggregations and complementarities⁴ (Supplementary Fig. 12–13). Supplementary Fig. 9 shows the change in extreme low-reliability power events during the past four decades. Unfortunately, extreme low-reliability events also followed a significant increasing trend over the past four decades. For example, duration of extreme low-reliability power shortage events over the periods between 1980–1990 and 1991–2000 have increased by 0.19.

Similar to extreme long-duration events, despite the global upwards trend in extreme low-reliability events, there are considerable discrepancies across different countries across the past four decades. For instance, the variability in frequency of extreme long-duration events ranged from nearly +13 in Benin between 1991–2000 and 2001–2010 to only around –12 in Suriname between 2001–2010 and 2011–2022 (Supplementary Fig. 9). However, most countries experienced growth in either frequency, duration, and/or intensity of extreme low-reliability events on a decadal scale (Supplementary Fig. 9). For example, there are 52% countries with growing intensity of low-reliability events during between 1980–1990 and 1991–2000. This case is even more evident between 2001–2010 and 2011–2022, for which 63% countries evinced uptrend in intensity of low-reliability events (Supplementary Fig. 9i).

Supplementary Figure 7 | Interannual variability in extreme long-duration events over the past 4 decades. **a-c**, Interannual changes in the frequency (**a**), duration (**b**), and intensity (**c**) of extreme long-duration events since 1980. The coefficients at the top of the panels indicate robust Theil-Sen's slopes and their corresponding P values, examined by Mann-Kendall (MK) test. *** represents the significance under the level of $P < 0.001$. The right boxplots denote the difference among the periods of 1980–1990, 1991–2000, 2001–2010, and 2011–2022. Black dashed lines denote linear fitting of annual average values across 178 countries. **d-f**, Rank ordering of the annual average frequency (**d**), duration (**e**), and intensity (**f**) of extreme long-duration events across the surveyed 43 years, for which the periods of 1980–1990, 1991–2000, 2001–2010, and 2011–2022 are coloured with deep blue, shallow blue, shallow red, and deep red, respectively.

Supplementary Figure 8 | Changes in extreme long-duration events over the past 4 decades. a-c, Changes in annual average frequency (a), duration (b), and intensity (c) of extreme long-duration events during the period 1980–1990 and 1991–2000. **d-f,** Changes in annual average frequency (d), duration (e), and intensity (f) of extreme long-duration events during the period 1991–2000 and 2001–2010. **g-i,** Changes in annual average frequency (g), duration (h), and intensity (i) of extreme long-duration events between 2001–2010 and 2011–2022. The right digits denote the share of countries with increasing long-duration events.

Supplementary Figure 9 | Changes in extreme low-reliability events over the past 4 decades. a-c, Changes in annual average frequency (a), duration (b), and intensity (c) of extreme low-reliability events during the period 1980–1990 and 1991–2000. **d-f,** Changes in annual average frequency (d), duration (e), and intensity (f) of extreme low-reliability events during the period 1991–2000 and 2001–2010. **g-i,** Changes in annual average frequency (g), duration (h), and intensity (i) of extreme low-reliability events between 2001–2010 and 2011–2022. The right digits denote the share of countries with increasing low-reliability events.

9. Other Questions on the manuscript

λ (1) The introduction has analyzed the research gap, but the challenge for the study is not so evident.

Response: We thank the Referee for the constructive comment. The knowledge gap of potential extreme shortage events may be the clear definition and systematic indicators, as well as the uniform analytical framework of climate-change-driven potential extreme shortage events. We have reorganized the Introduction (Lines 35–45) of revised manuscript and have added the potential challenges for this study.

Introduction

Although a few studies have provided valuable state-^{19,20}, national-^{21,22}, and regional-specific²³ insights on wind-solar supply shortfalls, the global understanding of the impact of climate change on extreme power shortage events of wind-solar hybrid systems remains largely unclear. However, as the previously limited attention to this emerging issue of electricity supply security, so far there has been no uniform analytical framework of climate-change-driven potential extreme shortage events, by combining macro projections of wind-solar generation shares (generally provided by integrated assessment models, IAMs) with hourly mismatches between wind-solar supply and electricity demand. Moreover, there has been still lack of a clear definition and systematic indicators of potential extreme power shortage events in wind-solar electricity systems, which in turn restricts our efforts towards energy security with growing renewable energies under changing climate.

λ (2) Figure 1 has only mapped the case of $P < 0.001$, while Figure 3 only mapped the case of $P < 0.05$. why? More details on the “ P ” are necessary.

Response: Sorry to this confusion. The P value describes the probability under the assumption of null hypothesis, which is applied to examine the statistical significance of observed interannual change (using Mann-Kendall test and ANOVA). In initial Figure 1, the frequency, duration, and intensity of long-duration events all have significantly increased under the significance of level of $P < 0.001$, thereby only mapping $P < 0.001$. Similarly, the frequency, duration, and intensity of low-reliability events all show an uptrend under the significance level of $P < 0.05$ in initial Figure 3, and thus we only labeled $P < 0.05$. In revised manuscript, we have reanalyzed the trends in potential power shortage events in 178 countries in Figure 1 and 3, and the resulting significance levels are under the level of $P < 0.001$ and $P < 0.01$, respectively.

In order to avoid potential confusion on statistical analysis and its significance level, we have added more details on statistical tests and its P value in *Method* section (Lines 445–455) of revised manuscript.

Method

Statistical analysis

We explored the linear trend in power shortage events, capacity factors, and climatic factors using Theil-Sen's slope estimator, a classical non-parametric method to detect interannual variation^{55,56}. We further applied Mann-Kendall test to examine the statistical significance of Theil-Sen's slope⁵⁷. The observed decadal changes in power shortage events, capacity factors, and climatic factors were examined using ANOVA that is a widely used method to detect statistically significant differences between the mean values of various groups⁵⁸. The P value, which describes the probability under assumption of null hypothesis^{56,57}, is applied to determine the significance of Mann-Kendall test and ANOVA. All the statistical analyses and significance detection were performed using R version (4.3.1) (R Foundation for Statistical Computing) with software packages.

Figure 1 | Interannual variability in extreme long-duration events for the surveyed 178 countries since 1980s. a-c, Interannual changes in the frequency (a), duration (b), and intensity (c) of extreme long-duration events since 1980. The coefficients at the top of the panels indicate robust Theil-Sen's slopes and their corresponding P values, examined by Mann-Kendall (MK) test. The right boxplots denote the difference before 2000 (blue) and after 2000 (red) examined by analysis of variance (ANOVA). *** represents the significance under the level of $P < 0.001$. Black dashed lines denote linear fitting of annual average values across surveyed 178 countries. **d-f,** Rank ordering of the annual average frequency (d), duration (e), and intensity (f) of extreme long-duration events across the surveyed 43 years, in which years between 1980 and 2000 are labelled with black and those between 2001 and 2022 are labelled with red.

Figure 3 | Changes in extreme low-reliability events since 1980. a-c, Changes in annual average frequency (a), duration (b), and intensity (c) of extreme low-reliability events between 1980–2000 and 2001–2022. The left-bottom boxplots denote the difference between 1980–2000 (blue) and 2000–2022 (red) examined by ANOVA, where ** represent the significance under the level of $P < 0.01$. **d,** Changes in extreme low-reliability events for 42 major countries between 1980–2000 and 2001–2022.

λ (3) It would be better to name the major economies in Figure 4, like China and UK.

Response: We thank the Referee for the thoughtful suggestion. We have named the major economies (e.g., China and UK) as suggested in Figure 4, and we have also checked all other figures and labelled the major economies in the same way (i.e., Figure 5 and Extended Figure 3).

Figure 4 | Relationship of changes in extreme long-duration and low-reliability events at the national scale. Changes in frequency (a), duration (b), and intensity (c) of extreme long-duration and low-reliability events for 42 major countries between 1980–2000 and 2001–2022. Red digits represent the proportion of countries that fall in the quadrant. Dashed red lines with shallow red shading are linear fittings between the two types of events and their 95% confidence intervals. SAU, MOZ and VEN denote Saudi Arabia, Mozambique, and Venezuela, respectively.

Figure 5 | Changes in extreme power shortage events with extremely low wind speed and solar radiation. a-c, Relationship between anomalies in frequency (a), duration (b), and intensity (c) of extreme power shortage events and anomalies in annual hours of extremely low solar radiation and wind speed at hub height for 42 major countries. d, Relationship between change in annual average wind speed and change in annual hours of extremely low wind speed between 1980–2000 and 2001–2022. e, Relationship between the relative change in annual hours of extremely low wind speed and the relative change in extreme power shortage events between 1980–2000 and 2001–2022. MOZ and VEN denote Mozambique and Venezuela, respectively. The lines and circles denote individual country’s change and the average changes across 42 major countries, respectively. The relative change above 100% is visualized as 100% change.

λ (4) It is necessary to improve the figures (for example, the texts were hidden by the figure body as in Extended Data Fig. 1 and Extended Data Fig. 2).

Response: We thank the Referee for the valuable comment. We have polished our figures as suggested in revised manuscript. In particular, the texts in Extended Data Fig. 1 and Extended Data Fig. 2 have been moved to avoid being hidden by the figure body.

Extended Data Fig. 1 | Schematic diagram of frequency, duration, and intensity of extreme power shortage events. There were four extreme long-duration events (frequency=4) in Germany in 1981, and one of these events occurred during the period from 15:30 on Jan. 26 to 08:30 on Feb. 1st (duration=138 hours), with the total power gap of 50.76 (intensity=50.76).

Extended Data Fig. 2 | Interannual variability in extreme long-duration events across different continents for the selected 178 countries. a-c, Annual average frequency (a), duration (b), and intensity (c) and their changes in extreme long-duration events at the continental scale between 1980–2000 and 2001–2022. The right column summarizes the two-decadal changes in extreme long-duration events for individual continents between 1980–2000 and 2001–2022.

Reviewer #2 (Remarks to the Author):

The manuscript investigates extreme power shortage events in wind-solar systems, utilizing 43 years of reanalysis climatological data from MERRA-2 across 42 countries. The authors aim to understand the dynamics and implications of these events on a national and global scale, drawing insights from these statistical analyses.

Response: We thank the Referee for the accurate summary of our work and for the fair and thoughtful comments below. We've made a number of substantial revisions in response, and believe the manuscript has been substantially improved. A point-to-point response is presented below.

Major Comments:

1. The manuscript's analysis appears to lack a robust design that sufficiently supports the conclusions drawn. The utilization of a single reanalysis product (MERRA-2) without a clear rationale or comparison with other available reanalysis datasets undermines the comprehensiveness and reliability of the findings. A more detailed justification for the exclusive selection of MERRA-2 or an incorporation of multiple reanalysis products, including ERA series, would strengthen the study's credibility.

Response: We thank the Referee for the constructive comment. We totally agree that multiple reanalysis products, like widely used ones such as MERRA-2 and ERA series datasets, can strengthen our findings. We have added ERA5 reanalysis product and related analysis as suggested during this round of revision.

The results show that, despite a slight underestimate in both long-duration and low-reliability power shortage events (Supplementary Fig. 1), our results based upon ERA5 data are largely in accordance with those using MERRA-2 (Supplementary Figs. 2–6). In belief, the results of ERA5 also showed an increasing trend in extreme long-duration events and low-reliability events across the globe (Supplementary Figs. 2–4), particularly in low- and middle-latitude developing countries (Supplementary Fig. 5). Such increasing trends in extreme power shortage events are largely ascribed to growing extremely low wind speed and solar radiation (Supplementary Fig. 6). We have added the sensitivity tests of ERA5 data in Results (Lines 90–92, 135–137, 161–163, and 225–228), Discussion (Lines 247–252) section and Supplementary Text 1 of revised manuscript.

Results

Variability in extreme long-duration shortage events

Such uptrend in extreme long-duration events was also captured by another ERA5 reanalysis data, despite potential underestimates compared to MERRA-2 reanalysis data (Supplementary Text 1 and Supplementary Figs. 1–6).

Change in extreme low-reliability shortage events

This upward trend in low-reliability events was confirmed by the estimates based upon ERA5 data (Supplementary Text 1 and Supplementary Fig. 4).

Relationship between extreme long-duration and low-reliability shortage events

The estimates based upon another ERA5 data also support the variability characteristics of two types of extreme power shortage events (Supplementary Text 1 and Supplementary Fig. 5).

Drivers of growing extreme shortage events

The disproportionated variabilities between extreme power shortage events and wind speeds was evidenced by the results using ERA5 reanalysis data (Supplementary Text 1 and Supplementary Fig. 6).

Discussion

On the other hand, the estimates of system reliability may be sensitive to reanalysis data choice⁴, therefore, we have added ERA5 data to verify our main results (Supplementary Text 1). It shows that the estimates based upon ERA5 data are largely in accordance with those using MERRA-2 (Supplementary Figs. 2–6), despite a slight underestimate in both long-duration and low-reliability power shortage events (Supplementary Fig. 1).

Supplementary Text 1. Sensitive tests of ERA5 reanalysis data on extreme power shortage events

Given the potential uncertainty of climatic variables from one-single weather data product (e.g., MERRA-2), it is necessary to verify our results using another independent reanalysis data. To address such a concern, we downloaded climatological data (including surface solar radiation downwards, top net solar radiation-clear sky, 100m u-component of wind speed, 100m v-component of wind speed, as well as 2m temperature) from ERA5 reanalysis product (<https://cds.climate.copernicus.eu/cdsapp#!/home>). We applied the same methods with MERRA-2 to calculate the hourly capacity factors of wind and solar power at a gridded scale, and then the gridded results were aggregated to estimate area-weighted average wind and solar capacity factors for individual countries. Afterwards, the resulting capacity factors were used to estimate system reliability and potential extreme power shortage events.

Supplementary Fig. 2 shows the characteristics of defined extreme long-duration events for wind-solar supply systems during the period 1980–2022 using ERA5 reanalysis data products. Globally, wind-solar supply systems have experienced an increasing trend in extreme long-duration events, although there are repeated up and down undulations irrespective of their frequency, duration, and intensity. Long-duration events worldwide increased significantly at a rate of 0.031 y^{-1} in frequency ($P < 0.001$; Supplementary Fig. 2a), 0.322 y^{-1} in duration (Supplementary Fig. 2b), and 0.120 y^{-1} in intensity (Supplementary Fig. 2c). In particular, we find that extreme long-duration events over the last two decades considerably outnumbered those over the first two decades. For example, the annual average duration of extreme long-

duration events rose evidently from 141.0 hours during 1980–2000 to 148.5 hours during 2001–2022 (one-way analysis of variance (ANOVA), $P < 0.001$; Supplementary Fig. 2b).

We further ranked the annual average frequency, duration, and intensity of extreme long-duration events over the past 43 years (Supplementary Fig. 2d–f). Our results revealed that the greatest frequency, duration, and intensity of extreme long-duration events occurred intensively in the period between 2001 and 2022. For instance, the period from 2001 to 2022 accounted for 85% of the top 20 frequent years of extreme long-duration events (Supplementary Fig. 2d). This case is even more apparent for the top 10 frequent years, 90% of which emerged over the period after 2000 (Supplementary Fig. 2d). Moreover, the highest years for the three metrics are different (i.e., 2022 for frequency, 2013 for duration, and 2020 for intensity), which in turn suggests potential complexity in dispatching flexible electricity sources to defend against extreme power shortage events.

Spatial analyses further illustrated that the observed uptrend in extreme long-duration events is a global-scale phenomenon (Supplementary Fig. 3). There are most countries with growing extreme long-duration events during the period of 1980–2000 and 2001–2022. In fact, there over two-third of surveyed 178 countries with uptrends in frequency, duration, intensity of extreme long-duration events, respectively (Supplementary Fig. 3). However, this ascending trend in extreme long-duration events at the global scale masks large spatial disparities across various countries. Indeed, the change in the frequency of extreme long-duration events peaked at +3.7 in Costa Rica, while bottoming at only –1.3 hours in Venezuela (Supplementary Fig. 3h).

Supplementary Fig. 4 shows the change in extreme low-reliability power events between 1980–2000 and 2001–2022 based upon ERA5 data. Unfortunately, extreme low-reliability events also followed a significant increasing trend over the past four decades. In the last two decades, extreme low-reliability events have increased by 1.0 (0.050 y^{-1}) in frequency (Supplementary Fig. 4a), 6.7 (0.253 y^{-1}) in duration (Supplementary Fig. 4b), and 2.3 (0.077 y^{-1}) in intensity (Supplementary Fig. 4c) relative to the counterpart in the first two decades ($P < 0.01$). Similar to extreme long-duration events, despite the global upwards trend in extreme low-reliability events, there are considerable discrepancies across different countries. For instance, the increase in the duration of extreme low-reliability events reached up to 124.9 hour in Indonesia, but only grew by 20.9 hours in Japan and even decreased by –3.4 hours in Tunisia (the second column in Supplementary Fig. 4d).

We further explored the relationship of the two types of extreme power shortage events for 178 countries, most of which presented the same trends for the two types of extreme power shortage events (Supplementary Fig. 5). Moreover, changes in extreme power shortage events are unevenly distributed but rather exhibit a latitudinal gradient and an evident disparity in countries with different economic development levels (Supplementary Fig. 5). Specifically, developed countries in high latitudes generally show slight variability in extreme power shortage events, yet developing countries in low and middle latitudes tend to have drastic changes in extreme power shortage events. For example, the greatest changes in frequency of extreme power shortage events have emerged in developing countries in low- and middle-

latitude countries (e.g., Nepal, Honduras and Costa Rica; blue bubbles at the edge of quadrants in Supplementary Fig. 5a); however, high-latitude countries generally exhibit a small variability in frequency of extreme power shortage events.

In addition, we revealed climatological drivers underlying the detected growing extreme power shortage events by combining extremely low wind speed and solar radiation using ERA5 data (Supplementary Fig. 6). We find that, the growth in extreme shortage events is largely attributable to prolonged extremely low wind speed (change in colors) rather than solar radiation (change in size) during the period between 1980–2000 and 2001–2022. In addition, there are large disproportionalities between changes in extreme power shortage events and variabilities in wind speeds. Small changes in annual average wind speed generally indicate considerable changes in extremely low wind speed, even resulting in a larger variability in extreme power shortage events. In fact, even though the change in annual average wind speed across the 178 countries is as low as 1.6% (the pink circle in Supplementary Fig. 6d), the annual hours of extremely low wind speed change by over 10.0% (the blue circle in Supplementary Fig. 6d and e). Moreover, such over 11.62% change in the annual hours of extremely low wind speed results in up to average 21.37% change in extreme long-duration and low-reliability power shortage events (the red circle in Supplementary Fig. 6e).

	Long-duration events						Low-reliability events					
	Frequency		Duration		Intenisty		Frequency		Duration		Intenisty	
	MERRA-2	ERA5	MERRA-2	ERA5	MERRA-2	ERA5	MERRA-2	ERA5	MERRA-2	ERA5	MERRA-2	ERA5
Annual mean	4.8	4.4	151.1	144.8	51.1	48.9	22.0	19.1	59.9	53.0	20.5	18.2
1980	4.6	4.1	147.3	145.2	49.4	48.5	23.2	19.9	51.2	41.9	17.8	14.6
1981	4.4	4.0	144.9	134.0	48.2	44.3	23.0	20.2	58.3	59.0	20.1	20.1
1982	4.7	4.3	155.7	148.4	52.4	50.0	22.7	20.2	62.0	57.4	21.2	19.5
1983	4.1	3.7	141.8	134.2	46.7	44.0	21.3	18.2	61.0	52.2	20.6	17.7
1984	4.5	4.0	144.1	140.4	48.4	47.4	22.8	20.4	57.3	53.1	19.5	18.1
1985	4.7	4.4	150.1	148.8	50.6	50.7	22.2	19.2	51.7	59.8	17.9	20.1
1986	4.3	3.9	140.3	136.6	47.9	46.9	21.0	18.0	51.0	47.9	17.6	16.5
1987	5.0	4.4	146.0	137.2	48.9	45.9	22.5	19.0	53.1	45.3	18.3	15.7
1988	4.4	4.0	146.3	142.2	49.0	47.8	20.3	18.1	51.8	47.5	18.0	16.4
1989	4.4	4.0	151.6	147.8	51.8	50.2	21.8	19.1	57.0	48.5	19.6	16.8
1990	4.3	3.9	150.0	141.6	50.4	47.8	20.4	17.5	63.7	54.4	21.7	18.7
1991	4.7	4.4	151.6	144.4	52.0	49.7	21.2	18.1	58.6	49.7	20.1	17.1
1992	4.6	4.2	142.1	134.8	47.7	45.4	20.7	17.5	51.9	45.5	17.9	15.7
1993	4.3	3.8	142.2	138.1	47.2	46.3	20.3	17.3	51.5	48.5	17.5	16.5
1994	4.3	4.1	139.8	128.9	46.5	43.4	20.7	18.0	55.9	49.9	19.2	17.0
1995	4.8	4.4	141.3	137.7	47.0	46.2	21.3	18.7	58.6	48.6	20.0	16.7
1996	4.8	4.3	147.2	141.4	49.6	48.1	21.5	18.7	49.6	47.2	17.3	16.3
1997	4.8	4.3	151.6	144.8	52.1	50.0	21.1	18.4	62.6	60.9	21.5	20.7
1998	4.6	4.2	146.9	145.9	49.8	49.2	19.6	17.9	63.6	58.4	21.7	19.8
1999	4.6	4.3	153.8	146.9	51.2	49.0	21.4	18.6	63.5	55.1	21.7	19.0
2000	4.6	4.1	151.2	140.9	51.3	47.4	21.0	18.1	50.7	44.3	17.6	15.4
2001	4.6	4.1	149.0	140.8	50.0	47.3	20.5	17.3	46.7	44.5	16.2	15.4
2002	4.3	3.8	143.1	140.9	48.6	47.9	20.4	17.8	48.1	44.5	16.8	15.5
2003	4.9	4.3	153.5	144.3	51.8	49.0	20.9	18.4	54.9	51.6	18.9	17.7
2004	4.5	4.1	150.9	149.2	51.5	51.5	21.1	18.7	59.0	55.4	20.2	18.8
2005	5.2	4.8	151.0	138.4	51.2	46.7	22.2	19.6	55.4	47.2	19.1	16.3
2006	5.3	5.0	159.3	152.8	54.3	52.2	22.0	19.7	60.5	50.8	20.8	17.6
2007	4.7	4.3	154.3	143.3	51.9	48.1	21.1	18.8	67.7	63.6	23.3	21.8
2008	5.4	5.1	163.7	156.3	55.7	53.5	22.6	20.2	62.3	54.0	21.5	18.7
2009	5.0	4.7	157.1	153.4	53.0	51.5	22.1	20.1	59.5	53.5	20.5	18.3
2010	5.7	5.3	154.6	147.5	53.0	50.4	24.4	21.6	69.3	60.4	23.8	20.7
2011	5.8	5.3	162.1	154.3	55.2	52.2	22.5	20.0	69.1	58.2	23.6	19.9
2012	5.3	4.8	152.6	147.6	51.2	49.9	23.4	20.1	59.2	63.0	20.6	21.5
2013	5.3	4.7	161.5	160.6	54.3	53.8	21.9	19.4	78.6	54.7	26.5	18.9
2014	5.3	4.7	154.9	148.5	52.6	50.0	23.2	20.9	64.3	57.2	22.0	19.5
2015	5.0	4.5	159.8	148.6	54.6	50.9	22.1	19.4	65.2	53.8	22.4	18.7
2016	5.5	5.0	153.1	145.1	52.2	49.2	23.5	21.2	74.2	55.6	24.9	19.0
2017	5.3	4.9	153.5	144.7	52.3	48.6	23.7	20.6	63.1	56.5	21.9	19.4
2018	4.8	4.3	159.3	147.8	53.7	49.8	23.6	21.0	65.2	64.3	22.4	21.9
2019	5.2	4.6	151.9	145.9	51.0	48.7	23.3	20.5	54.9	49.2	19.0	17.1
2020	5.3	4.7	155.0	154.9	53.7	53.9	22.8	19.5	74.3	55.2	25.2	19.1
2021	5.2	4.4	154.1	146.0	51.1	47.6	23.4	18.4	67.6	58.1	23.2	19.7
2022	5.5	4.8	157.7	155.7	54.2	53.0	23.4	18.8	70.4	51.0	24.1	17.7

Supplementary Figure 1 | Comparison of interannual variability in extreme power shortage events between MERRA-2 and ERA5 data. Shading in each block represents annual average power shortage events during the period of 1980–2022, with the colour range from red (high values) to blue (low values).

Supplementary Figure 2 | Interannual variability in extreme long-duration events since 1980s using ERA5 data. a-c, Interannual changes in the frequency (a), duration (b), and intensity (c) of extreme long-duration events since 1980. The coefficients at the top of the panels indicate robust Theil-Sen's slopes and their corresponding *P* values, examined by Mann-Kendall (MK) test. The right boxplots denote the difference before 2000 (blue) and after 2000 (red) examined by analysis of variance (ANOVA). *** represents the significance under the level of $P < 0.001$. Black dashed lines denote linear fitting of annual average values across surveyed 178 countries. d-f, Rank ordering of the annual average frequency (d), duration (e), and intensity (f) of extreme long-duration events across the surveyed 43 years, in which years between 1980 and 2000 are labelled with black and those between 2001 and 2022 are labelled with red.

Supplementary Figure 3 | Maps of extreme long-duration events during the period 1980–2000 and 2001–2022 using ERA5 data. a-c, Annual average frequency (a), duration (b), and intensity (c) of extreme long-duration events during the period 1980–2000. **d-f,** Annual average frequency (d), duration (e), and intensity (f) of extreme long-duration events during the period 2001–2022. **g-i,** Changes in annual average frequency (g), duration (h), and intensity (i) of extreme long-duration events between 1980–2000 and 2001–2022.

Supplementary Figure 4 | Changes in extreme low-reliability events since 1980 using ERA5 data. a-c, Changes in annual average frequency (a), duration (b), and intensity (c) of extreme low-reliability events between 1980–2000 and 2001–2022. The left-bottom boxplots denote the difference between 1980–2000 (blue) and 2001–2022 (red) examined by ANOVA, where * and ** represent the significance under the level of $P < 0.05$ and $P < 0.01$, respectively. **d,** Changes in extreme low-reliability events for 42 major countries between 1980–2000 and 2001–2022.

Supplementary Figure 5 | Relationship of changes in extreme long-duration and low-reliability events at the national scale calculated through ERA5 data. Changes in frequency (a), duration (b), and intensity (c) of extreme long-duration and low-reliability events for 178 surveyed countries between 1980–2000 and 2001–2022. Red digits represent the proportion of countries that fall in the quadrant. Dashed red lines with shallow red shading are linear fittings between the two types of events and their 95% confidence intervals. D. Congo denotes Democratic Republic of the Congo.

Supplementary Figure 6 | Changes in extreme power shortage events with extremely low wind speed and solar radiation using ERA5 data. a-c, Relationship between anomalies in frequency (a),

duration (b), and intensity (c) of extreme power shortage events and anomalies in annual hours of extremely low solar radiation and wind speed at hub height for individual countries. d, Relationship between change in annual average wind speed and change in annual hours of extremely low wind speed between 1980–2000 and 2001–2022. e, Relationship between the relative change in annual hours of extremely low wind speed and the relative change in extreme power shortage events between 1980–2000 and 2001–2022. The relative change above 100% is visualized as 100% change. The lines and circles denote individual country's change and the average changes across 178 surveyed countries, respectively. The surrounding digits represent the order number of the 178 countries listed in Supplementary Table 6.

2. Similarly, the choice of 42 'major' (according to the authors) countries for analysis is not adequately justified. The criteria for selecting these specific countries, and their representativeness, are not clearly articulated. A more transparent criterion for country selection, possibly based on regional representation or other relevant factors, is essential to validate the study's conclusions at national, regional, and global levels.

Response: We thank the Referee for the valuable comment. A more transparent criterion for country selection in the main text has been added as suggested. The 42 "major" countries were selected based on the factors of both electricity demand and regional representation. To be specific, we selected the top 10 countries with the greatest electricity demand from four continents (i.e., Asia, Europe, Africa, and America), apart from Oceania in which only two main countries (e.g., Australia and New Zealand) were chosen (Supplementary Table 3). The resulting 42 major countries represented ~87% of the total power demand around the world.

Additionally, we have added new analysis for 178 countries (i.e., available power demand) that cover ~99% of power demand across the globe (Supplementary Table 1). The expanded results from the 178 countries do not alter our conclusions (see revised Results section, new Figs. 1-3, Extended Figs. 2 and 4, as well as Supplementary Figs. 19 and 20). Meanwhile, given that visualizing too many countries may overshadow information about major economies that the scientific community usually pays great attentions, we still focus on the characteristic analysis of 42 major countries in revised Figs. 4 and 5. Meanwhile, we also present the corresponding analyses for 178 countries in Supplementary Text 4 and Supplementary Figs. 14 and 15.

Supplementary Text 4. Selection criteria of 42 major countries and the expanded results of the whole 178 countries

As visualizing too many countries may overshadow information about major economies that the scientific community usually pays great attentions, we present some results for 42 major countries (Figs. 4–5). These countries were chosen based on power demand and regional representation. To be specific, we selected the top 10 countries with the highest electricity demand for individual continents (e.g., Asia, Europe, Africa, and America), apart from Oceania in which only two countries (e.g., Australia and New Zealand) were chosen (Supplementary Table 2). The resulting 42 major countries represented ~87% of the total power demand around

the world¹⁶. Our study also shows the corresponding results (Supplementary Figs. 14 and 15) across the global 178 countries (Supplementary Table 1) that covers ~99% of power demand across the globe¹⁶, and the expanded results do not alter our main conclusions.

As shown by Supplementary Fig. 14, most of which presented the same trends for the two types of extreme power shortage events across the global 178 countries. Indeed, there are approximately 61% countries where trends in the frequency of extreme long-duration events are in line with those of extreme low-reliability events (upper-right and bottom-left quadrants in Supplementary Fig. 14). By contrast, the change in extreme low-reliability events is not entirely in accordance with that in extreme long-duration events. For example, Georgia exhibited a growth in frequency of extreme long-duration events (+0.80) but a decrease in frequency of extreme low-reliability events (-4.48).

Moreover, changes in extreme power shortage events are unevenly distributed but rather exhibit a latitudinal gradient and an evident disparity in countries with different economic development levels (Supplementary Fig. 14). Specifically, developed countries in high latitudes generally show slight variability in extreme power shortage events, yet developing countries in low and middle latitudes tend to have drastic changes in extreme power shortage events. For example, the greatest changes in frequency of extreme power shortage events have emerged in developing countries in low- and middle-latitude countries (e.g., Nepal, Honduras and Costa Rica; blue bubbles at the edge of quadrants in Supplementary Fig. 14a); however, high-latitude countries generally exhibit a small variability in frequency of extreme power shortage events.

Supplementary Fig. 15 shows climatological drivers underlying the detected growing extreme power shortage events by combining extremely low wind speed and solar radiation across the whole 178 countries. We find that, the growth in extreme shortage events is largely attributable to prolonged extremely low wind speed (change in bubble's colors) rather than solar radiation (change in bubble's size) during the period between 1980–2000 and 2001–2022. In addition, there are large disproportionalities between changes in extreme power shortage events and variabilities in wind speeds. Small changes in annual average wind speed generally indicate considerable changes in extremely low wind speed, even resulting in a larger variability in extreme power shortage events. In fact, even though the change in annual average wind speed across the 178 countries is as low as 0.65% (the pink circle in Supplementary Fig. 15d), the annual hours of extremely low wind speed change by over 11.62% (the blue circle in Supplementary Fig. 20d and e). Moreover, such over 11.62% change in the annual hours of extremely low wind speed results in up to average 21.39% change in extreme long-duration and low-reliability power shortage events (the red circle in Supplementary Fig. 15e).

Figure 1 | Interannual variability in extreme long-duration events for the surveyed 178 countries since 1980s. a-c, Interannual changes in the frequency (a), duration (b), and intensity (c) of extreme long-duration events since 1980. The coefficients at the top of the panels indicate robust Theil-Sen's slopes and their corresponding *P* values, examined by Mann-Kendall (MK) test. The right boxplots denote the difference before 2000 (blue) and after 2000 (red) examined by analysis of variance (ANOVA). *** represents the significance under the level of $P < 0.001$. Black dashed lines denote linear fitting of annual average values across surveyed 178 countries. **d-f,** Rank ordering of the annual average frequency (d), duration (e), and intensity (f) of extreme long-duration events across the surveyed 43 years, in which years between 1980 and 2000 are labelled with black and those between 2001 and 2022 are labelled with red.

Figure 2 | Maps of extreme long-duration events for the surveyed 178 countries during the period 1980–2000 and 2001–2022. a-c, Annual average frequency (a), duration (b), and intensity (c) of extreme long-duration events during the period 1980–2000. d-f, Annual average frequency (d), duration (e), and intensity (f) of extreme long-duration events during the period 2001–2022. g-i, Changes in annual average frequency (g), duration (h), and intensity (i) of extreme long-duration events between 1980–2000 and 2001–2022.

Figure 3 | Changes in extreme low-reliability events since 1980. a-c, Changes in annual average frequency (a), duration (b), and intensity (c) of extreme low-reliability events between 1980–2000 and 2001–2022. The left-bottom boxplots denote the difference between 1980–2000 (blue) and 2001–2022 (red) examined by ANOVA, where ** represent the significance under the level of $P < 0.01$. d, Changes in extreme low-reliability events for 42 major countries

between 1980–2000 and 2001–2022.

Supplementary Figure 14 | Relationship of changes in extreme long-duration and low-reliability events for the selected 178 countries. Changes in frequency (a), duration (b), and intensity (c) of extreme long-duration and low-reliability events for 178 surveyed countries between 1980–2000 and 2001–2022. Red digits represent the proportion of countries that fall in the quadrant. Dashed red lines with shallow red shading are linear fittings between the two types of events and their 95% confidence intervals.

Supplementary Figure 15 | Changes in extreme power shortage events with extremely low wind speed and solar radiation for the selected 178 countries. a-c, Relationship between anomalies in frequency (a), duration (b), and intensity (c) of extreme power shortage events and anomalies in annual hours of extremely low solar radiation and wind speed at hub height for individual countries. d, Relationship between change in annual average wind speed and change in annual hours of extremely low wind speed between 1980–2000 and 2001–2022. e, Relationship between the relative change in annual hours of extremely low wind speed and the relative change in extreme power shortage events between 1980–2000 and 2001–2022. The relative change above 100% is visualized as 100% change. The lines and circles denote individual country's change and the average changes across 178 surveyed countries, respectively. The surrounding digits represent the order number of the 178 countries that are listed in Supplementary Table 6.

Extended Data Fig. 2 | Interannual variability in extreme long-duration events across different continents for the selected 178 countries. a-c, Annual average frequency (a), duration (b), and intensity (c) and their changes in extreme long-duration events at the continental scale between 1980–2000 and 2001–2022. The right column summarizes the two-decadal changes in extreme long-duration events for individual continents between 1980–2000 and 2001–2022.

Extended Data Fig. 4 | Changes in extreme power shortage events with extremely low wind speed and solar radiation worldwide for the surveyed 178 countries. a-c, Changes in frequency (a), duration (b), and intensity (c) of extreme long-duration events with annual hours of extremely low wind speed and solar radiation. d-f, Changes in frequency (d), duration (e), and intensity (f) of extreme low-reliability events with annual hours of extremely low wind speed and solar radiation.

Supplementary Table | 1 The selected 178 major countries/regions

Continents	Involved countries
Oceania	New Zealand, Australia, Kiribati, New Caledonia, Papua New Guinea, Solomon Islands, Vanuatu
Asia	China, India, Japan, South Korea, Indonesia, Saudi Arabia, Iran, Thailand, Vietnam, Pakistan, Malaysia, Philippines, Kazakhstan, United Arab Emirates, Singapore, Bangladesh, Iraq, Uzbekistan, Israel, Kuwait, Hong Kong, Myanmar, Hungary, Qatar, Syria, Sri Lanka, Oman, Bahrain, Jordan, Lebanon, Turkmenistan, Tajikistan, North Korea, Yemen, Cameroon, Kyrgyzstan, Nepal, Afghanistan, Armenia, Laos, Mongolia, Brunei, Bhutan

Europe	France, United Kingdom, Italy, Poland, Turkey, Russia, Spain, Germany, Sweden, Ukraine, Norway, Netherlands, Belgium, Finland, Czech Republic, Austria, Romania, Switzerland, Greece, Portugal, Belarus, Bulgaria, Kosovo, Serbia, Ethiopia, Azerbaijan, Denmark, Ireland, Slovakia, Guatemala, Iceland, Bosnia and Herzegovina, Slovenia, Lithuania, Macedonia, Moldova, Latvia, Luxembourg, Democratic Republic of the Congo, Montenegro, Malta
America	United States, Brazil, Mexico, Argentina, Canada, Venezuela, Colombia, Peru, Chile, Ecuador, Cuba, Puerto Rico, Croatia, Uruguay, Costa Rica, Paraguay, Panama, Bolivia, Trinidad and Tobago, Honduras, Nicaragua, Cyprus, Gabon, Haiti, Jamaica, Guinea, Bahamas, French Guiana, Suriname, Belize, Cape Verde, Antigua and Barbuda
Africa	Egypt, Algeria, Ghana, South Africa, Tunisia, Nigeria, Mozambique, Libya, Sudan, Morocco, Angola, South Sudan, Dominican Republic, Burkina Faso, Kenya, Tanzania, Zambia, Uganda, Zimbabwe, El Salvador, Georgia, Ivory Coast, Albania, Niger, Cambodia, Estonia, Togo, Botswana, Namibia, Chad, Senegal, Rwanda, Mauritius, Mali, Madagascar, Republic of Congo, Malawi, Benin, Somalia, Equatorial Guinea, Mauritania, Western Sahara, Eritrea, Sierra Leone, Swaziland, Guinea-Bissau, Fiji, Guyana, Liberia, Burundi, Lesotho, Djibouti, Gambia, Central African Republic, Seychelles, Sao Tome and Principe

Supplementary Table | 3 The selected 42 major countries across the world

Continents	Involved countries
Oceania	New Zealand, Australia
Asia	China, India, Indonesia, Iran, Japan, Malaysia, Saudi Arabia, Thailand, South Korea, Vietnam
Europe	Turkey, Russia, France, Germany, Italy, Sweden, Ukraine, United Kingdom, Spain, Poland
America	United States, Canada, Brazil, Mexico, Venezuela, Argentina, Chile, Colombia, Paraguay, Peru
Africa	Algeria, Egypt, Ghana, Libya, Morocco, Mozambique, Nigeria, South Africa, Sudan, Tunisia

3. A refined methodological approach, considering additional reanalysis products and a broader or more systematic country selection, is recommended to bolster the study's credibility.

Response: We appreciate the Referee for these great comments. Both an additional reanalysis product and a broader country selection have been added as suggested to bolster the study's credibility.

On one hand, additional reanalysis product of ERA5 has been included to conduct all the simulations. The results show that our estimates of extreme power shortage events based upon ERA5 data are largely in accordance with those using MERRA-2 (see above Supplementary Text 1 and Supplementary Figs. 1–6).

On the other hand, other 136 countries and regions have been added (totally 178 countries and regions) for broader analysis. The expanded results from the 178 countries do not alter our conclusions and even show a more significant increasing trend in extreme power shortage events (see above Fig. 1–3, Supplementary Text 4, Supplementary Figs. 14 and 15, Extended Figure 2 and 4).

4. While I have several minor comments, I believe addressing the fundamental issues related to research design and methodology first is paramount.

Response: We greatly appreciate the Referee for constructive comments. We have made substantial revisions to improve the design and methodology of this study as suggested, and will also carefully respond other minor comments. Thanks for helping the continuous improvement of our manuscript.

REVIEWER COMMENTS

Reviewer #1 (Remarks to the Author):

After reading the author's response to the first round of comments and revisions, I consider that the amendments required were all well implemented. However, there are some issues to be addressed additionally.

1. This study highlights extremely low wind speed may be responsible for the observed growing extreme power shortage events. However, it's essential to note that the installed capacities of wind and solar in various countries may not aligned with the "simplified" planning. Moreover, nearly all countries are wind-heavy (Supplementary Fig. 39), indicating a greater proportion of power from wind resources. Therefore, the impact of wind droughts on power shortage is naturally more significant. Given the assumption made in this study, the conclusion is nearly inevitable. Thus, what practical significance does this conclusion hold?

2. Although the study did not isolate the contributions of extreme-low wind speed and solar radiation to extreme power shortage events, it concluded that an average 8.80% change in extremely low wind speed gives rise to over 30% variability in extreme power shortage events. This conclusion lacks sufficient persuasiveness.

3. In examining the climatological drivers of growing extreme power shortage events, this study focused solely on the individual variations in wind and solar resources but overlooked situations of compound drought involving wind and solar resources. This may be a significant factor contributing to power shortage events and warrants further investigation.

4. The title of this study is "Climate change impacts on the extreme power shortage events of wind-solar supply systems worldwide". However, the study only analyzed data from 1980 to 2022 in several decades, without considering future climate change scenarios. The title should be revised to avoid potential misunderstandings.

Reviewer #2 (Remarks to the Author):

Dear authors,

I would like to express my appreciation for the detailed and thoughtful revisions you made to your manuscript in response to the reviewers' comments, including mine. Your effort has significantly improved the paper, and it's clear that you've addressed the feedback with great dedication.

Congratulations on your hard work, and I look forward to seeing your study contribute to our field.

Reviewer #1 (Remarks to the Author):

After reading the author's response to the first round of comments and revisions, I consider that the amendments required were all well implemented. However, there are some issues to be addressed additionally.

Response: We thank the Referee for satisfying with our first round of revisions. And we have made a number of revisions in response to address the following issues, and believe the manuscript has been further improved. A point-to-point response is presented below.

1. This study highlights extremely low wind speed may be responsible for the observed growing extreme power shortage events. However, it's essential to note that the installed capacities of wind and solar in various countries may not aligned with the “simplified” planning. Moreover, nearly all countries are wind-heavy (Supplementary Fig. 39), indicating a greater proportion of power from wind resources. Therefore, the impact of wind droughts on power shortage is naturally more significant. Given the assumption made in this study, the conclusion is nearly inevitable. Thus, what practical significance does this conclusion hold?

Response: We thank the Referee for this constructive comment. We totally agree that potential uncertainties from wind and solar installed capacities and wind/solar ratio may affect our estimates in extreme power shortage events and their driving forces. Our initial conclusion is based on a wind-heavy electricity system from the reliability-optimized method. We have supplemented two additional sensitivity tests based upon integrated assessment models (IAMs) as suggested.

First, we have added a sensitivity test of wind and solar installed capacities predicted by IAMs (Byers et al., 2022; <https://data.ece.iiasa.ac.at/ar6/>) to examine the potential impact of installed capacities. The results showed that the two types of power shortage events also exhibit a growing trend since 1980 (Supplementary Fig. 20), with the uptrend in extremely low wind speed and extremely low solar radiation (Supplementary Figs.21–22). Although the impact of extremely low wind speed on the frequency of power shortage events is more significant, extremely low wind speed and solar radiation shows a nearly identical contribution to the duration and intensity of power shortage events (Supplementary Fig. 22).

Second, in view of the uncertainty from the proportion of wind and solar power, we have conducted an additional sensitivity test of wind/solar ratio according to the projection of IAMs by this mid-century (International Institute for Applied Systems Analysis (IIASA); <https://data.ece.iiasa.ac.at/ar6/>; Supplementary Table 6). As shown in Supplementary Table 10, wind power constitutes 39.8–79.6% of wind-solar mix supply systems, with an average of 48.0% across the world (i.e., a greater solar penetration). The resulting extreme power shortage events have increased between 1980 and 2022 (Supplementary Fig. 23). The observed uptrends in extreme power shortage events are also associated with growing extremely low wind speed and solar radiation (Supplementary Fig. 24), but indeed wind speed does not always place a dominant position in extreme power shortage events (Supplementary Fig. 25).

In summary, the contribution of wind and solar power to extreme shortage events would be sensitive to wind/solar ratio and installed capacities. Therefore, together with other suggestions from the Referee, we define a new index combining wind and solar drought (e.g., compound extremely low wind speed and solar radiation events, a period over which both wind speed and solar radiation are below the 10th percentile of the daily average value across 43 surveyed years). Then, we have conducted additional analyses to underscore the influence of compound extremely low wind speed and solar radiation events to extreme shortage events, which would in turn inform energy security management and planning of wind-solar supply systems.

Finally, we have revised the description texts about the driving forces of power shortage events, and have added the above two sensitivity tests in revised Abstract (Lines 11–14), Introduction (Lines 75–80), Results (Lines 191–201), Discussion (Lines 264–270), as well as Supplementary Texts 5–6.

Abstract

This uptrend in extreme power shortage events is driven by extremely low wind speed and solar radiation, particularly compound wind and solar drought, which however are strongly disproportionated.

Introduction

Additionally, a series of sensitivity tests on potential constraints such as reanalysis products (i.e., ERA5), solar supply technologies (i.e., single-axis solar tracking system and concentrating solar power, CSP), IAMs-based wind-solar ratio and installed capacities, the total wind-solar supply shares (predicted by REMIND model; Supplementary Table 4)^{26,27}, and decadal analyses (i.e., 1980–1990, 1991–2000, 2001–2010, and 2011–2022) were performed to investigate their impacts on extreme power shortage events.

Results

We find that, the growth in extreme shortage events is attributable to prolonged extremely low wind speed and solar radiation (Fig. 5a–c), which also conforms with significant declines in the annual mean wind speed and solar radiation since the 1980s ($P < 0.05$; Supplementary Fig. 16–19 and Supplementary Text 3). Although variabilities in extreme power shortage events are controlled by both wind and solar energies, their relative importance are different and are partly affected by wind-solar ratio and installed capacities. The changes in extreme power shortage events are primarily ascribed by the variability in wind power in wind-heavy system (Extended Data Fig. 4), while additional analyses using IAMs-based wind/solar ratio (solar-heavy system) and install capacities reveal that wind power does not always place a dominant position in extreme power shortage events (Supplementary Texts 5–6 and Supplementary Figs. 20–25).

Discussion

Our study may also be affected by wind-solar supply parameter settings and demand profile. To be specific, we estimated wind-solar ratio and installed capacities based on the reliability-optimized method, which likely in turn influences our results about power shortage events and their driving forces. Our extended sensitivity tests with IAMs-based wind and solar proportion and install capacities do not alter the uptrend in power shortage events, while these tests indicate that the importance of wind and solar energies would differ from various extreme shortage events and their metrics (Supplementary Texts 5–6 and Supplementary Figs. 20–25).

Supplementary text 5. Sensitive tests of sensitivity test of wind and solar installed capacities based on the IAMs prediction

Given the potential impact of wind and solar installed capacities on extreme power shortage events and their drivers, we collected wind and solar installed capacities across the world predicted by IAMs in this mid-century from the International Institute for Applied Systems Analysis (IIASA)¹⁶ (<https://data.ece.iiasa.ac.at/ar6/>). The wind and solar installed capacities for individual countries' have been estimated based upon electricity demand and the global installed capacities predicted by IAMs. We applied the IAMs-based installed capacities of wind and solar power to examine the variabilities in extreme power shortage events and their driving forces.

Supplementary Fig. 20 shows the characteristics of defined extreme long-duration and low-reliability events for wind-solar supply systems during the period 1980–2022 using IAMs-based wind and solar installed capacities. Globally, wind-solar supply systems have experienced an increasing trend in extreme long-duration and low-reliability events, although there are repeated up and down undulations irrespective of their frequency, duration, and intensity. Indeed, long-duration events worldwide increased significantly at a rate of 0.012 y^{-1} in frequency ($P < 0.001$; Supplementary Fig. 20a), 0.451 y^{-1} in duration (Supplementary Fig. 20b), and 0.181 y^{-1} in intensity (Supplementary Fig. 20c). In particular, we find that extreme long-duration events over the last two decades considerably outnumbered those over the first two decades. For example, the annual average duration of extreme low-reliability events rose evidently from 26.97 hours during 1980–2000 to 28.61 hours during 2001–2022 (one-way analysis of variance (ANOVA), $P < 0.001$; Supplementary Fig. 20e).

We further revealed climatological drivers underlying the detected growing extreme power shortage events by combining extremely low wind speed and solar radiation (Supplementary Fig. 21). We find that, the growth in extreme shortage events is largely attributable to prolonged extremely low wind speed (change in colors) and extremely low solar radiation (change in size) during the period between 1980–2000 and 2001–2022. Although variabilities in extreme power shortage events are controlled by both wind and solar drought, their importance differ across different metrics. The impact of wind droughts on the frequency of power shortage events is more significant, while wind drought and solar drought show a nearly

identical contribution to the duration and intensity of power shortage events (Supplementary Fig. 22).

Supplementary text 6. Sensitive tests of sensitivity test of wind/solar ratio based on the IAMs prediction

In light of the potential uncertainty of wind/solar proportion towards extreme power shortage events and their driving forces, we acquired the supply share of wind and solar power according to the projections of IAMs by this mid-century from the International Institute for Applied Systems Analysis (IIASA) (<https://data.ece.iiasa.ac.at/ar6/>; Supplementary Table 6). As shown in Supplementary Table 6, wind power constitutes 39.8–79.6% of wind-solar mix supply systems, with an average of 48.0% across the world.

As shown by Supplementary Fig. 23, wind-solar supply systems have experienced an increasing trend in extreme long-duration and low-reliability events, although there are repeated up and down undulations irrespective of their frequency, duration, and intensity. In fact, long-duration events worldwide increased significantly at a rate of 0.015 y^{-1} in frequency ($P < 0.001$; Supplementary Fig. 23a), 0.426 y^{-1} in duration (Supplementary Fig. 23b), and 0.177 y^{-1} in intensity (Supplementary Fig. 23c). In particular, we find that extreme long-duration events over the last two decades considerably outnumbered those over the first two decades. For example, the annual average frequency of extreme low-reliability events increased evidently from 35.33 hours during 1980–2000 to 37.98 hours during 2001–2022 (one-way analysis of variance (ANOVA), $P < 0.001$; Supplementary Fig. 23d).

Supplementary Figure 24 shows that potential climatological drivers underlying the detected growing extreme power shortage events by combining extremely low wind speed and solar radiation. We find that, the growth in extreme shortage events is largely attributable to prolonged extremely low wind speed (change in colors) and extremely low solar radiation (change in size) during the period between 1980–2000 and 2001–2022. Although variabilities in extreme power shortage events are controlled by both wind and solar drought, their importance differ across different extreme events and their metrics (Supplementary Fig. 25). For example, the impact of wind droughts on the frequency of long-duration shortage events is more significant (Supplementary Fig. 25a), while solar droughts largely dominate the variabilities in the duration and intensity of long-duration shortage events (Supplementary Fig. 25b–c).

Supplementary Figure 20 | Interannual variability of extreme long-duration events and low-reliability events estimated based upon wind and solar installed capacities predicted by IAMs. a-c, Interannual changes in the frequency (a), duration (b), and intensity (c) of extreme long-duration events since 1980. **d-f,** Interannual changes in the frequency (d), duration (e), and intensity (f) of extreme low-reliability events since 1980. The coefficients at the top of the panels indicate robust Theil-Sen's slopes and their corresponding P values, examined by Mann-Kendall (MK) test. The right boxplots denote the difference before 2000 (blue) and after 2000 (red) examined by analysis of variance (ANOVA). *** and ** represent the significance under the level of $P < 0.001$ and $P < 0.01$, respectively. Black dashed lines denote linear fitting of annual average values of extreme power shortage events.

Supplementary Figure 21 | Changes in extreme power shortage events with extremely low wind speed and solar radiation based on wind and solar installed capacities predicted by IAMs. a-c, Relationship between anomalies in frequency (a), duration (b), and intensity (c) of extreme power shortage events and anomalies in annual hours of extremely low solar radiation and wind speed at hub height for individual countries.

Supplementary Figure 22 | Changes in extreme power shortage events with extremely low wind speed and solar radiation worldwide based upon wind and solar installed capacities predicted by IAMs. a-c, Changes in frequency (a), duration (b), and intensity (c) of extreme long-duration events with annual hours of extremely low wind speed and solar radiation. **d-f,** Changes in frequency (d), duration (e), and intensity (f) of extreme low-reliability events with annual hours of extremely low wind speed and solar radiation.

Supplementary Figure 23 | Interannual variability of extreme long-duration events and low-reliability events estimated based upon wind/solar ratio predicted by IAMs. **a-c**, Interannual changes in the frequency (**a**), duration (**b**), and intensity (**c**) of extreme long-duration events since 1980. **d-f**, Interannual changes in the frequency (**d**), duration (**e**), and intensity (**f**) of extreme low-reliability events since 1980. The coefficients at the top of the panels indicate robust Theil-Sen's slopes and their corresponding P values, examined by Mann-Kendall (MK) test. The right boxplots denote the difference before 2000 (blue) and after 2000 (red) examined by analysis of variance (ANOVA). *** and ** represent the significance under the level of $P < 0.001$ and $P < 0.01$, respectively. Black dashed lines denote linear fitting of annual average values of extreme power shortage events.

Supplementary Figure 24 | Changes in extreme power shortage events with extremely low wind speed and solar radiation based upon wind/solar ratio predicted by IAMs. a-c, Relationship between anomalies in frequency (a), duration (b), and intensity (c) of extreme power shortage events and anomalies in annual hours of extremely low solar radiation and wind speed at hub height for individual countries.

Supplementary Figure 25 | Changes in extreme power shortage events with extremely low wind speed and solar radiation worldwide based upon wind/solar ratio predicted by IAMs. a-c, Changes in frequency (a), duration (b), and intensity (c) of extreme long-duration events with annual hours of extremely low wind speed and solar radiation. d-f, Changes in frequency (d), duration (e), and intensity (f) of extreme low-reliability events with annual hours of extremely low wind speed and solar radiation.

Supplementary Table 6 | Predicted wind and solar shares to the total wind-solar supply systems by mid-century based on IAMs

Geographical zones	Wind share	Solar share
World	48.04%	51.96%
Africa	39.78%	60.22%
South Africa	50.62%	49.38%
Asia	45.83%	54.17%
China	47.37%	52.63%
India	45.22%	54.78%
Indonesia	70.64%	29.36%
Japan	66.48%	33.52%
Pakistan	45.30%	54.70%
South Korea	55.59%	44.41%
Europe	58.24%	41.76%
Turkey	56.47%	43.53%
Middle East	42.30%	57.70%
North America	45.89%	54.11%
Canada	66.43%	33.57%
Mexico	55.64%	44.36%
United States	46.76%	53.24%
Oceania	64.76%	35.24%
Eurasia	45.83%	54.17%
Russia	59.09%	40.91%
Azerbaijan	58.24%	41.76%
Central and South America	45.89%	54.11%
Brazil	66.16%	33.84%

Reference:

Byers, E., Krey, V., Kriegler, E. & Riahi, K. AR6 Scenarios Database hosted by IIASA. Int. Inst. Appl. Syst. Anal. (2022) doi:10.5281/zenodo.5886911.\

2. Although the study did not isolate the contributions of extreme-low wind speed and solar radiation to extreme power shortage events, it concluded that an average 8.80% change in extremely low wind speed gives rise to over 30% variability in extreme power shortage events. This conclusion lacks sufficient persuasiveness.

Response: We appreciate the Referee for the constructive comment. We are sorry for making an inaccurate description. This initial conclusion means the period between 1980–2000 and 2001–2022 have experienced 8.80% increase in annual average hours of extremely low wind speed and 30.38% growth in extreme power shortage events. Thus, we concluded that an average 8.80% change in extremely low wind speed may give rise to over 30% variability in extreme power shortage events. In order to avoid potential confusion, we combine the resource droughts of both wind and solar to design a new index instead, compound extremely low wind

speed and solar radiation events. It is defined as a period, during which both wind speed and solar radiation fall below the 10th percentile of the daily average value across 43 surveyed years (Richardson et al., 2023; Richardson et al., 2022; Rinaldi et al., 2021).

We find that, the annual hours of compound extremely low wind speed and solar radiation have increased across the world between 1980–2000 and 2001–2022 (Extended Data Fig. 5). The growing annual average hours of compound extremely low wind speed and solar radiation events are associated with the uptrends in both long-duration and low-reliability extreme power shortage events since 1980 (Supplementary Fig. 26). Moreover, we have compared the change in average wind speed and average solar radiation, compound extremely low wind speed and solar radiation events, as well as extreme power shortage events. The results showed that a comparatively small change in average wind speed and average solar radiation usually suggest a large variability in compound extremely low wind speed and solar radiation event, thus even leading a much higher change in extreme power shortage events (Figs. 5d and e).

We have added such analyses of compound extremely low wind speed and solar radiation events in revised Abstract (Lines 11–17), Results (Lines 202–232), and Supplementary Text 7.

Abstract

This uptrend in extreme power shortage events is driven by extremely low wind speed and solar radiation, particularly compound wind and solar drought, which however are strongly disproportionated. Only average 12.5% change in compound extremely low wind speed and solar radiation events may give rise to over 30% variability in extreme power shortage events, despite a mere average 1.0% change in average wind speed and average solar radiation.

Results

Moreover, we further define a compound extremely wind speed and solar radiation event, a period during which both wind speed and solar radiation are below the 10th percentile of the daily average value across 43 surveyed years^{20,31,32}. We find that most countries have experienced increasing defined compound extremely low wind speed and solar radiation. Indeed, there over 70% countries with an uptrend in annual hours of compound extremely low wind speed and solar radiation events, although some countries (e.g., Peru, Algeria, and France) exhibit the decline in compound extremely low wind speed and solar radiation between 1980–2000 and 2001–2022 (Supplementary Text 7 and Extended Data Fig. 5). More importantly, such a growing trend in compound extremely low wind speed and solar radiation events is in largely accordance with the observed increase in extreme long-duration and low-reliability shortage events since the 1980s (Supplementary Fig. 26). For instance, higher intensity of long-duration events in the recent period (e.g., 2010, 2016, and 2022) have been in line with more hours of compound extremely low wind speed or solar radiation events, while a lower intensity of long-duration events before 2000 (e.g., 1983 and 1998) is consistent with a shorter compound extremely low wind speed or solar radiation.

Notably, there are large disproportionalities between changes in extreme power shortage events and climatological variables (e.g., compound extremely wind speed and solar radiation events

and average wind speed and solar radiation; Fig. 5d–e). Small changes in annual average wind speed and average solar radiation generally indicate considerable changes in compound extremely wind speed and solar radiation events. Indeed, although the change in annual average wind speed and average solar radiation is as low as 1.0% (the pink circle in Fig. 5d), the annual hours of compound extremely low wind speed and solar events change by more than 12.5% (the blue circle in Fig. 5d–e). More importantly, small changes in compound extremely low wind speed and solar events usually suggest substantial changes in extreme power shortage events. In fact, a mere 12.5% change in the annual hours of compound extremely low wind speed and solar events would drives an average 30.4% change in extreme long-duration and low-reliability events (the red circle in Fig. 5e). The results were evidenced by the results using ERA5 reanalysis data (Supplementary Text 1 and Supplementary Fig. 6). The disproportionalities of changes between extreme power shortage events and climatological variables highlight that even mild climate changes are expected to pose a severe challenge to the security of wind-solar generation systems.

Supplementary text 7. Expanded analyses of compound extremely low wind speed and solar radiation events and power shortage events

In order to disentangle the potential driving forces behind the growing power shortage events, we define compound extremely low wind speed and solar radiation events, during which both wind speed and solar radiation are below the 10th percentile of the daily average value across 43 surveyed years. The results showed that the annual hours of defined compound extremely low wind speed and solar radiation during 2001–2022 (500.89 hours on average) was evidently higher than those during 1980–2000 (528.84 hours on average) across the world (Extended Data Fig. 5).

Spatial analyses further illustrated that the observed uptrend in annual hours of defined compound extremely low wind speed and solar radiation is a global-scale phenomenon (Extended Data Fig. 5). There are most countries with growing extreme long-duration events during the period of 1980–2000 and 2001–2022. In fact, there over 70% of surveyed 178 countries with uptrend in annual hours of defined compound extremely low wind speed and solar radiation, although some countries located in North Africa, South America, and Europe evinced a decreasing trend in annual hours of defined compound extremely low wind speed and solar radiation. Moreover, the change in compound extremely low wind speed and solar radiation events also shows a latitude gradient (Extended Data Fig. 5). High-latitude countries (e.g., Canada and Sweden) usually have smaller variabilities in compound extremely low wind speed and solar radiation, while countries located in low latitudes (such as India and Mozambique) tend to favor larger changes in compound extremely low wind speed and solar radiation.

Supplementary Fig. 26 shows that the relationship between changes in extreme power shortage events and variabilities in annual hours of compound extremely low wind speed and solar radiation events. The increases in extreme power shortage events are in accordance with

significant growth in annual hours of compound extremely low wind speed and solar radiation events since the 1980s. For instance, highly frequent extreme power shortage events in the recent period (e.g., 2022, 2016, 2010, and 2013) have resulted from comparatively long annual hours of compound extremely low wind speed or solar radiation events, while a lower frequency of extreme power shortage events before 2000 (e.g., 1983 and 1994) is ascribed to shorter periods of compound extremely low wind speed or solar radiation events.

Moreover, we have compared the change in average wind speed and solar radiation, compound extremely low wind speed and solar radiation events, as well as extreme power shortage events during the period of 1980–2000 and 2001–2022 (Figure 5d–e). The results showed that a comparatively small change in average wind speed and solar radiation usually suggest a large variability in compound extremely low wind speed and solar radiation event. Indeed, the change in average wind speed and average solar radiation of 1.0% (the pink circle in Fig. 5d) imply 12.5% variability in annual hours compound extremely low wind speed and solar radiation events (the blue circle in Fig. 5d–e), which further triggers off over 30% change in extreme power shortage events (the red circle in Fig. 5e).

Extended Data Fig. 5 | Changes in annual hours of compound extremely low wind speed and solar radiation events for individual countries between 1980–2000 and 2001–2022. The left-bottom boxplots denote the difference of compound extremely low wind speed and solar radiation events for individual countries between 1980–2000 (blue) and 2000–2022 (red) examined by ANOVA, where *** represent the significance under the level of $P < 0.001$.

Supplementary Figure 26 | Relationship between extreme power shortage events and compound extremely low wind speed and solar radiation. a-c, Relationship between annual average frequency (a), duration (b), and intensity (c) of extreme power shortage events and annual average hours of compound extremely low wind speed and solar radiation between 1980 and 2022.

Figure 5 | Changes in extreme power shortage events with extremely low wind speed and solar radiation. a-c, Relationship between anomalies in frequency (a), duration (b), and intensity (c) of extreme power shortage events and anomalies in annual hours of extremely low solar radiation and wind speed at hub height for 42 major countries. d, Relationship between change in annual average wind speed and change in annual hours of compound extremely low wind speed & solar radiation between 1980–2000 and 2001–2022. e, Relationship between the relative change in annual hours of extremely low wind speed & solar radiation and the relative change in extreme power shortage events between 1980–2000 and 2001–2022. MOZ and VEN denote Mozambique and Venezuela, respectively. The lines and circles denote individual country’s change and the average changes across 42 major countries, respectively.

References:

Richardson, D., Pitman, A. J. & Ridder, N. N. Climate influence on compound solar and wind droughts in Australia. *npj Clim. Atmos. Sci.* 6, 1–10 (2023).

Richardson, D. et al. Global increase in wildfire potential from compound fire weather and drought. *npj Clim. Atmos. Sci.* 5, (2022).

Rinaldi, K. Z., Dowling, J. A., Ruggles, T. H., Caldeira, K. & Lewis, N. S. Wind and solar resource droughts in California highlight the benefits of long-term storage and integration with the western interconnect. *Environ. Sci. Technol.* 55, 6214–6226 (2021).

3. In examining the climatological drivers of growing extreme power shortage events, this study focused solely on the individual variations in wind and solar resources but overlooked situations of compound drought involving wind and solar resources. This may be a significant factor contributing to power shortage events and warrants further investigation.

Response: We thank the Referee for this meaningful suggestion. We totally agree that compound drought involving wind and solar resource would be a significant factor contributing to extreme power shortage events. As described above, we define compound extremely low wind speed and solar radiation events as suggested, during which both wind speed and solar radiation fall below the 10th percentile of the daily average value across 43 surveyed years (Richardson et al., 2023; Richardson et al., 2022; Rinaldi et al., 2021). Then, we disentangle the trend in compound extremely low wind speed and solar radiation events and its associations with extreme power shortage events. The detailed descriptions have been described in the response to Comment #2 to avoid repetition, and we have added these revisions in Abstract (Lines 11–17), Results (Lines 202–232), and Supplementary Text 7 as previously mentioned.

Additionally, we have also modified the previous corresponding sensitivity analyses of Fig. 5d–e to analyze the relationship between compound extremely low wind speed and solar radiation events and extreme power shortage events (See Supplementary Texts 1, 4, 10, and 11; Supplementary Figs. 6, 15, 36, and 43).

Supplementary Text 1. Sensitive tests of ERA5 reanalysis data on extreme power shortage events

In addition, we revealed climatological drivers underlying the detected growing extreme power shortage events by combining extremely low wind speed and solar radiation using ERA5 data (Supplementary Fig. 6). We find that, the growth in extreme shortage events is largely attributable to prolonged extremely low wind speed (change in colors) and solar radiation (change in size) during the period between 1980–2000 and 2001–2022. In addition, there are large disproportionalities between changes in extreme power shortage events and variabilities in compound extremely low wind speed and solar radiation events. Small changes in annual

average wind speed and solar radiation generally indicate considerable changes in annual hours of compound extremely low wind speed and solar radiation events, even resulting in a larger variability in extreme power shortage events. In fact, even though the change in annual average wind speed and solar radiation across the 178 countries is as low as 1.62% (the pink circle in Supplementary Fig. 6d), change in annual hours of compound extremely low wind speed and solar radiation events reach 9.55% (the blue circle in Supplementary Fig. 6d and e). Moreover, such nearly 10% change in compound extremely low wind speed and solar radiation events results in up to average 21.37% change in in extreme long-duration and low-reliability power shortage events (the red circle in Supplementary Fig. 6e).

Supplementary Text 4. Selection criteria of 42 major countries and the expanded results of the whole 178 countries

Supplementary Fig. 15 shows climatological drivers underlying the detected growing extreme power shortage events by combining extremely low wind speed and solar radiation across the whole 178 countries. We find that, the growth in extreme shortage events is largely attributable to prolonged extremely low wind speed (change in bubble's colors) and solar radiation (change in bubble's size) during the period between 1980–2000 and 2001–2022. In addition, there are large disproportionalities between changes in extreme power shortage events and variabilities in wind speed and solar radiation. Small changes in annual average wind speed and solar radiation generally indicate considerable changes in compound extremely low wind speed and solar radiation, even resulting in a larger variability in extreme power shortage events. In fact, even though the change in annual average wind speed and solar radiation across the 178 countries is as low as 1.20% (the pink circle in Supplementary Fig. 15d), change in annual hours of extremely low wind speed and solar radiation events reached 13.62%% (the blue circle in Supplementary Fig. 20d and e). Moreover, such 13.62% change in the annual hours of extremely low wind speed results in up to average 21.39% change in in extreme long-duration and low-reliability power shortage events (the red circle in Supplementary Fig. 15e).

Supplementary Text 10. Sensitive tests of power shortage events with consideration of concentrating solar power technology (CSP)

Besides, we further investigate the relationship between changes in extreme events and variabilities in climatic variables, which still supports our finding that uptrend in compound extremely low wind speed and solar radiation events is behind the growing extreme power shortage events (Supplementary Fig. 36).

Supplementary Text 11. Sensitive tests of economics and renewable policy through using the wind-solar share predicted by REMIND model

Furthermore, we explored the potential mechanisms underlying such phenomenon, which also highlights compound extremely low wind speed and solar radiation events may be responsible for the observed growing extreme shortage events (see Supplementary Fig. 43).

Supplementary Figure 6 | Changes in extreme power shortage events with extremely low wind speed and solar radiation using ERA5 data. a-c, Relationship between anomalies in frequency (a), duration (b), and intensity (c) of extreme power shortage events and anomalies in annual hours of extremely low solar radiation and wind speed at hub height for the 178 surveyed countries. **d,** Relationship between change in annual average wind speed and change in annual hours of compound extremely low wind speed & solar radiation between 1980–2000 and 2001–2022. **e,** Relationship between the relative change in annual hours of extremely low wind speed & solar radiation and the relative change in extreme power shortage events between 1980–2000 and 2001–2022. The relative change above 100% is visualized as 100% change. The lines and circles denote individual country’s change and the average changes across 178 surveyed countries, respectively. The surrounding digits represent the order number of the 178 countries that are listed in Supplementary Table 7.

Supplementary Figure 15 | Changes in extreme power shortage events with extremely low wind speed and solar radiation for the selected 178 countries. **a-c**, Relationship between anomalies in frequency (**a**), duration (**b**), and intensity (**c**) of extreme power shortage events and anomalies in annual hours of extremely low solar radiation and wind speed at hub height for the 178 surveyed countries. **d**, Relationship between change in annual average wind speed and change in annual hours of compound extremely low wind speed & solar radiation between 1980–2000 and 2001–2022. **e**, Relationship between the relative change in annual hours of extremely low wind speed & solar radiation and the relative change in extreme power shortage events between 1980–2000 and 2001–2022. The relative change above 100% is visualized as 100% change. The lines and circles denote individual country's change and the average changes across 178 surveyed countries, respectively. The surrounding digits represent the order number of the 178 countries that are listed in Supplementary Table 7.

Supplementary Figure 36 | Changes in extreme power shortage events with extremely low wind speed and solar radiation with consideration of CSP technology. a-c, Relationship between anomalies in frequency (a), duration (b), and intensity (c) of extreme power shortage events and anomalies in annual hours of extremely low solar radiation and wind speed at hub height for the 178 surveyed countries. **d,** Relationship between change in annual average wind speed and change in annual hours of compound extremely low wind speed & solar radiation between 1980–2000 and 2001–2022. **e,** Relationship between the relative change in annual hours of extremely low wind speed & solar radiation and the relative change in extreme power shortage events between 1980–2000 and 2001–2022. The relative change above 100% is visualized as 100% change. The lines and circles denote individual country’s change and the average changes across 178 surveyed countries, respectively. The surrounding digits represent the order number of the 178 countries that are listed in Supplementary Table 7.

Supplementary Figure 43 | Changes in extreme power shortage events with extremely low wind speed and solar radiation using the wind-solar shares in REMIND mode. a-c, Relationship between anomalies in frequency (a), duration (b), and intensity (c) of extreme power shortage events and anomalies in annual hours of extremely low solar radiation and wind speed at hub height for the 178 surveyed countries. **d,** Relationship between change in annual average wind speed and change in annual hours of compound extremely low wind speed & solar radiation between 1980–2000 and 2001–2022. **e,** Relationship between the relative change in annual hours of extremely low wind speed & solar radiation and the relative change in extreme power shortage events between 1980–2000 and 2001–2022. The relative change above 100% is visualized as 100% change. The lines and circles denote individual country’s change and the average changes across 178 surveyed countries, respectively. The surrounding digits represent the order number of the 178 countries that are listed in Supplementary Table 7.

References:

Richardson, D., Pitman, A. J. & Ridder, N. N. Climate influence on compound solar and wind droughts in Australia. *npj Clim. Atmos. Sci.* 6, 1–10 (2023).

Richardson, D. et al. Global increase in wildfire potential from compound fire weather and drought. *npj Clim. Atmos. Sci.* 5, (2022).

Rinaldi, K. Z., Dowling, J. A., Ruggles, T. H., Caldeira, K. & Lewis, N. S. Wind and solar resource droughts in California highlight the benefits of long-term storage and integration with the western interconnect. *Environ. Sci. Technol.* 55, 6214–6226 (2021).

4. The title of this study is "Climate change impacts on the extreme power shortage events of wind-solar supply systems worldwide". However, the study only analyzed data from 1980 to 2022 in several decades, without considering future climate change scenarios. The title should be revised to avoid potential misunderstandings.

Response: Thanks for this valuable suggestion, and we are sorry for this inaccurate title that possibly allows potential confusion about the study period of this work. We have changed our title as "*Climate change impacts on the power shortage events of wind-solar supply systems worldwide during 1980–2022*" to avoid such a misunderstanding.

Reviewer #2 (Remarks to the Author):

Dear authors,

I would like to express my appreciation for the detailed and thoughtful revisions you made to your manuscript in response to the reviewers' comments, including mine. Your effort has significantly improved the paper, and it's clear that you've addressed the feedback with great dedication. Congratulations on your hard work, and I look forward to seeing your study contribute to our field.

Best regards,

Response: We appreciate that the Referee is satisfied with our revisions, and we also thank the important contributions of the Referee to this study.

REVIEWERS' COMMENTS

Reviewer #1 (Remarks to the Author):

I have no further questions, all my concerns and suggestions are carefully addressed.